# Universal Mobility in Old Core Cities of India:
## *People's Perception*

**Gaurab Das Mahapatra \*** , **Suguru Mori and Rie Nomura**

Laboratory of Architectural Planning, N216, Division of Architectural and Structural Design, Engineering Faculty, Hokkaido University, Kita 13-Jo, Nishi 8-Chome, Kita Ku, Sapporo 060-8628, Japan; suguru-m@eng.hokudai.ac.jp (S.M.); nomurarie@eng.hokudai.ac.jp (R.N.)
\* Correspondence: gaurabdasmahapatra@gmail.com; Tel.: +81-807-025-6958

**Abstract:** In this research, users' perception towards Universal Mobility in old core cities of India has been critically analyzed. Despite Universal Design guidelines from the United Nations and Union Government of India, old cities in India seldom have Universal Mobility, in effect endangering the lifestyle of senior citizens and differently-abled people. The core of Kolkata Municipal Corporation in Kolkata, India, has been considered a case example for this research. This research has considered three types of datasets for analysis. First, the authors interviewed 310 respondents from the Indian design fraternity, with the objective of understanding their opinions on the concept of Universal Design. In the next investigative study of 125 respondents from different wards of Kolkata Municipal Corporation, the purpose was to comprehend people's perception regarding walkability and mobility in an old Indian city. In the last visual survey of a stretch in Central Kolkata, the focus was on identifying hindrances in Universal Mobility in an old city core of Indian origin. Significant dissatisfaction was found regarding walkability amongst all user groups, which is linked to poor infrastructural conditions. Furthermore, accessing public transportation is difficult due to improper waiting facilities. However, the design fraternity in India suggests the need of separate accessibility guidelines for old and new cities in India. The design fraternity also recommends a customized rating system for accessing Universal Design. The result of this study indicates a need of recognizing the difficulty in imparting Universal Mobility in old core cities in India. This information can be used for preparing an access audit checklist through Architectural Planning, which is the first step in proposing a framework for Universal Mobility in old core cities in India.

**Keywords:** Universal Design; mobility; old core cities; walkability; Central Kolkata; Architectural Planning

## 1. Introduction

According to the 2011 Census of India, 2.21% of the Indian population, or 26.8 million people, are disabled [1]. The medically approved disability categories as per the 2011 Census were (a) vision, (b) auditory, (c) verbal, (d) movement, (e) mental retardation, (f) mental illness, (g) multiple disabilities, and (h) any disabilities other than those mentioned but clinically verified. Furthermore, only the people under the aforementioned medically approved categories were considered for the facilities/benefits provided to differently-abled people, as per the Indian guidelines such as "Article 41 of Constitution of India" or "The Person with Disabilities (PwD) Act, 1995" [2]. Article 41 of Indian constitution states that "The State shall, within the limits of its economic capacity and development, make effective provision for securing the right to work, to education and to public assistance in cases of unemployment, old age, sickness and disablement, and in other cases of undeserved want". The PwD Act, 1995 enlisted seven conditions of disabilities, namely: (a) blindness, (b) low vision, (c) leprosy cured, (d) hearing impairment, (e) locomotive disability, (f) mental retardation, and (g) mental illness. In contrast, the "Right to Persons with Disabilities Act,

2016" recognizes twenty-one types of health conditions (Ministry of Law and Justice, 2016). Thus, the number of "medically disabled" in India is substantially higher than the data published in the 2011 Census. In addition to this, the elderly people who might have been facilitated from the same provisions (as of differently-abled) were never considered as a stakeholder. As a result, the 103.8 million elderly people (as per the 2011 Census) were also not included in the category of receiving facility/benefit [3]. One of the hindrances in the Indian disability scenario is the national reluctance towards the shifting of focus from the medical model of disability towards a logical model.

"United Nations Convention on the Rights of Persons with Disabilities" (hereafter, UN-CRPD) and its Optional Protocol (A/RES/61/106) were adopted on 13 December 2006 at the United Nations Headquarters in New York, with an aim towards changing global attitudes and approaches towards persons with disabilities. Article 9 of the UN-CRPD suggests equal opportunities for differently-abled people in the following three aspects: (a) physical environment, (b) transportation, and (c) information and communication. Article 9 of the UN-CRPD also directs the member nations to implement "universally designed" public facilities [4]. Along these lines, Goal Number 11 of the "United Nations Sustainable Development Goals" (hereafter, the UN-SDG) specifies "Sustainable Cities and Communities" which aims towards making cities and human settlements inclusive, safe, resilient, and sustainable. Target 11.2 (within Goal Number 11 of the UN-SDG) elaborates that by 2030, all the member nations of the United Nations should provide access to safe, affordable, accessible, and sustainable transport systems for all. Additionally, the member nations should improve road safety, especially by expanding accessible public transport. Furthermore, goal 11 specifies that the aforementioned developments should be carried out with special/distinctive attention to the needs of (a) those in vulnerable situations, (b) women, (c) children, (d) persons with disabilities, and (e) older persons [5]. According to the data from World Population Prospects (the 2019 Revision), one in six people (16%) in the world will be over age 65 (16%) by 2050; in comparison to the ratio of one in eleven people (9%) in 2019. The main factors behind this phenomenon are declining fertility, increasing longevity, and international migration. Thus, in these times of globally changing demography, both the UN-CRPD and UN-SDG suggest the concept of Universal Mobility for enabling movement within a city without discrimination on the basis of physical or mental limitations. India, being a member nation, is required to act on these similar lines of action towards an "Universally Designed" built environment.

In contrast, most of the Indian documents related to Universal Design/accessible design are mere guidelines and not mandatory rules; thus, their implementation for accomplishing "barrier-free" or "inclusive" built environments at the societal and/or urban-level has not happened [6]. The authors further argue that in spite of being a member nation of the UN and one of the eighty-two signatories in UN-CRPD, India has a gap in the policy framework of Universal Design's implementation. However, the authors also suggest rectifying the gap by strategic interpretation of the latest disability data and taking into account the needs of Indian citizens. In India, the accessibility conditions are comparatively low in "old cities" in comparison to accessibility provisions in newly planned cities. However, in the Indian subcontinent, old cities can have multiple definitions. For this research, cities evolved during the early 18th century (start of British Era) are referred to as "old cities". As of Census 2011, India's population is over 1210 million, with a decadal growth rate of over 17%. Additionally, India's density increased from 325 person/ sq.km in 2001 to 382 people/sq.km in 2011 (with over 17.5% increase). The core areas in respective cities are further denser. In the case of Kolkata Municipal Corporation, the density is over 24,200 persons per sq. km. against a national figure of just 382 persons per sq. km. Owing to the ever-increasing population of India and high density in its historically-developed organically-planned core cities, "accessibility" becomes more complex than most of the countries abroad. "Accessibility" refers to the provisions for people including both able-bodied and differently-abled for accessing the urban facilities without discrimination. Thus, dealing with accessibility in old city parts of India is an interesting as well as an important

domain of urban infrastructure [7]. As an example, cities planned after independence of India (from British in 1947) such as Chandigarh and Bhubaneswar have relatively better facilities/options for differently-abled and senior citizens (people above 60 years), than cities such as old parts of imperial Kolkata or colonial Delhi.

As a general practice in urban India, the practices of Universal Design guidelines or Barrier-Free are comparatively more at Building level, and not at site/precinct level [8]. Universal Mobility connects the missing links between "Universally Designed" buildings and "Universally Designed" premises/precincts and creates accessible urban spaces. In India, a project such as "Mass Rapid Transit System" in Delhi (operationalized in 2002) has initiated the process of inclusive transportation [9]. However, on a large scale, the UN-CRPD's focus on Universal Mobility is not yet addressed in Indian guidelines. Likewise, Indian cities are still lagging behind in creating accessible urban spaces because of practicing "Inclusive" or "Accessible" Transportation", instead of "Universal Mobility". "Inclusive Transportation" focuses on making the mode of transport accessible to all, for example, accessible railway stations or accessible bus stops. The aspects covered in "Accessible Transportation" are (a) access to the station, (b) fare payment, (c) travelling information and communication, and (d) interior conditions of the mode of transport [10]. "Universal Mobility", in contrast, is a policy level intervention at the city scale ensuring a minimum standard of mobility for all members of society [11]. Thus, "Universal Mobility" is technically sound than "Inclusive "or "Accessible" Transportation".

Urban reforms in India particularly focusing on improving the quality of life in an inclusive way has substantially increased in recent years. One of such initiatives has been the "Accessible India Campaign" (launched in December 2015 by Ministry of Social Justice and Empowerment, Government of India) in consonance with Article 9 of UN-CRPD. "Accessible India Campaign" has a number of components which promote accessibility in (a) built environments, (b) transportation systems, and information and communication eco-systems [12].

In a similar line, through interviews and surveys during this research, the authors inferred that for this research, "Transportation System Accessibility" (involving vehicular traffic and walkability) be dealt with greater focus than the other two components. At an objective level, the focus of UN-CRPD towards Universal Mobility relates to the "Transportation System Accessibility" component of the Accessible India Campaign. The global focus and national demand made the authors focus on the topic of Universal Mobility rather than any other aspect of Universal Design for this research.

### 1.1. Research Components

This section specifies the following for this paper: (a) aims, (b) objectives, (c) research questions, (d) limitations, and (e) hypothesis.

The aim of this paper is to determine the status of accessibility in the core areas of an old Indian city by identifying factors for an ideal accessibility audit checklist. An accessibility audit checklist is an audit format to check the condition of spatial accessibility based on parameters (such as the presence of ramps) and their related indicators (such as slopes or material of the ramp).

The objectives to strengthen the aim are the following: (a) to ascertain the need for a new dimension in the Indian accessibility scenario, (b) to assess people's perspective towards Universal Mobility in core urban areas in India, and (c) to identify the issues in mobility in core urban areas in India.

The underlying research question for this paper is to find out whether core areas of urban India can be made inclusive in terms of accessibility. However, the research shall be limited to core urban areas in India, and the study shall be street level.

The hypothesis considered for this research is that the core cities in the Indian context need to be made accessible through provisions in planning and design. The paper shall only explore the factors that are required to constitute the parameters in an Accessibility Audit Checklist.

## 1.2. Research Process

This research paper began with the literature study (explained later in detail in Section 2 of this paper). At the same time, participatory audiences for the research were contacted. For substantiating Objective 1 (as mentioned in Section 1.1), the authors interacted with the design fraternity in India, and 310 people among those responded through questionnaires. For exploring Objective 2 (as mentioned in Section 1.1), the authors interviewed 125 residents from Kolkata Municipal Corporation (hereafter KMC). KMC is the municipal authority for the city of Kolkata which covers an area of 206.08 sq. km. with a population over 4.5 million and density of over 24,200 persons per sq. km. KMC's history dates back to 1726, during its formation by a royal charter from British Government. As on January 2021, KMC has jurisdiction of 144 wards (smallest administrative unit in Indian urban administrative system). For achieving Objective 3 (as mentioned in Section 1.1), a stretch of nearly 850 m within KMC limit was visually surveyed by the authors.

Summarizing fundamental theories and explanations of experts from the field of Universal Design and Accessibility was the next step in the research process. This section was further subdivided into three parts according to the genre of the documents referred: (a) Universal Design Theory, (b) Fundamental Understanding of Accessibility, (c) Universal Mobility in Urban Areas. This part is further elaborated in Section 2.

After the aforementioned stage, "choice of research" was determined by the authors for generating research content through interviews, field study, and field surveys. Additionally, content analysis was also done based on the research method. The method used for this research by the authors was "Critical Instance Case Study". "Critical Instance Case Study" methods studies are used to examine situations of unique interest, or to challenge an universal or generalized belief. Such studies are not focussed to create new generalizations. Rather, several situations or events may be examined to raise questions or challenge previously held assertions. Case Study type was suitable for this research because this paper required a detailed analysis of a delineated zone in core of old cities in India. The specificity of the study area made the authors prefer the "case study" method of research. Additionally, this research is aimed towards examining situations of Universal Mobility and thus defies the generalized format of mobility planning existing in old core Indian cities. This research critically appraises various opinions and scenarios to criticize mobility conditions in old core Indian cities. Thus, "Critical Instance" sub-type of "Case Study Research" deemed fit for this paper. The following stage was operationalization of the research.

The two components of operationalization were (a) measuring the data or creating measurable attributes for the data and (b) designing an appropriate questionnaire. The authors adopted the Likert scale approach and percentage of opinions for solving the first component (i.e., measuring the data or creating measurable attributes for the data). Three types of variables are used in this research: (a) categorical, (b) discrete, and (c) ranked. Categorical variables are used in a research when the variables can be compartmentalized into certain categories/groups. In this paper, categorical variables are used in the study of factors such as type of disability and type of infrastructure to be mapped. Discrete variables are used when there is a limitation or specific value to the variables. In this research, discrete variables are used in factors such as number of streets. Ranked variables are used when the available data set can be put in a sequence or order. In this paper, ranked Variables are used in factors involving opinions such as people's opinion on comfort level in walkability. For the second component (i.e., designing an appropriate questionnaire), the authors compared the questionnaire from various published research with the feedback from respondents during a pilot survey, before the actual questionnaire survey. This part is further explained in Section 3.1.1, Section 3.2.1, and Section 3.3.1 of this paper. Sampling strategy was the next step in the research.

Sampling strategy for this research was determined by feasibility study. The authors conducted three types of surveys to assess the feasibility of the project. First, the 310 samples for taking opinion of the design fraternity of India were linked to the number of participants who took part in the questionnaire survey provided by the authors at the end

of certain digital interactions (explained further in Section 3.1). Second, the 125 samples for understanding people's opinion were collected by the authors from different wards on KMC, with majority of respondents from in and around the central core of Kolkata (explained further in Section 3.2). Third, the visual observation study was conducted by the authors at an 850 m long stretch in the core of Kolkata. This selected stretch in the core of Kolkata has mixed land use, historic origin, and heavy traffic movement (Explained further in Section 3.3).

The authors observed and analyzed the collected data with an aim of "coding" or interpreting. This part is further explained in Section 3.1.2, Section 3.2.2, and Section 3.3.2 of this paper. While interpreting the data, the focus was on identifying the factors affecting Universal Mobility in the core of old Indian cities.

The following step for this paper addressed the major findings of the research through data manipulation. The authors considered the following points in this section: (a) co-relating the survey findings and (c) linking the objectives of the research to the data interpretation. This part is further explained in Section 4 of this paper.

The last part of the research process was finding out how this research could be fed into further research. This part is further explained in Section 5 of this paper. In this section, authors have mentioned the reason for the gap between existing policies and on-ground implementation of Universal Design guidelines in India. Subsequently, the need of an ideal "accessibility audit format" is also mentioned. Further, the role of "Laboratory of Architectural Planning, Hokkaido University" in the field of Universal Design and Accessibility is also mentioned briefly by the authors. Lastly, the authors explain how this research is just a beginning towards addressing the entire issue of accessibility in the core of old Indian cities. Additionally, a methodology for the scope of further research based on the findings from this paper is furnished by the authors.

Figure 1 summarizes the process undertaken for this research. The column on the left of Figure 1 shows the research stages and the column on the right explains the activity related to that research stage.

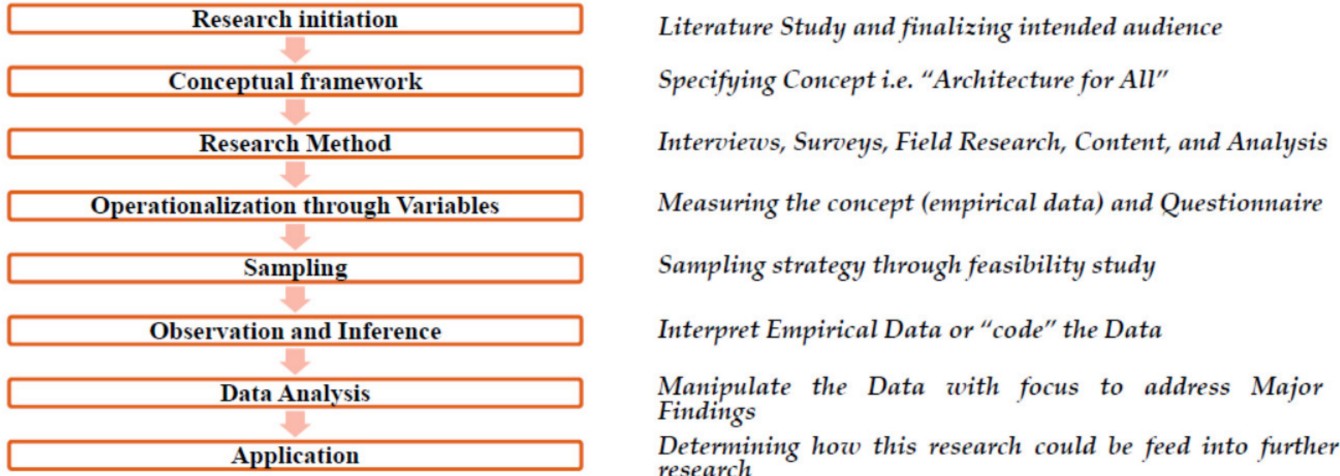

**Figure 1.** Research process followed for this paper (Source: Author).

## 2. Literature

Architect Ronald Mace, who coined the term Universal Design, defines Universal Design as the design of products and environments which is maximally usable by all people without the need for any adaptation or specialized design [13]. The "International Classification of Functioning, Disability and Health" of the "World Health Organization" (or, W.H.O.) mentions that disability is a phase, and it is not specific to any age or gender [14]. "Inclusive design features" which focusses on disabled friendly provisions in products and designed space within the built environment, affects the independence of

elderly/differently-abled to a great extent. Additionally, "exclusive" barrier-free mobility options at the city level (such as ramps and braille signage) improve the quality of life for the elderly/differently-abled. Conversely, hedonic adaptation (tendency of people to return to normal state after the occurrence of an extreme positive/negative event and age cohort effects (higher state of satisfaction due to the lower expectation) can undermine the need for such "exclusive" inclusive features [15]. Thus, the authors further state that instead of focusing on specific facilities for differently-abled and elderly people, encouraging "Universal Mobility" uplifts the inclusiveness of urban space.

## 2.1. Universal Design Theory

Goldsmith (2000) establishes the need for a bottom-up route towards Universal Design and proposes a model called "Universal Design Pyramid". Goldsmith's "Universal Design Pyramid" indicates that Universal Design facilitates the able-bodied and differently-abled alike [16]. Universal Design implementation on a national or regional-scale depends on the pattern in which user-experience based inputs are implemented into the planning legislation [17]. Likewise, Universal Design solutions are socially and financially rewarding since they make spaces easier for everyone to use [18]. Based on the inclusive civil engineering guidelines like IS 4963- Recommendations for Building and Facilities for the physically handicapped (1968), or Guidelines and Space standards for Barrier Free Built Environment for Disabled and Elderly persons (1998), or IRC 103-Guidelines for Pedestrian Facilities (2012), designers in India often provide separate provisions for differently-abled people; without realizing that separate facilities promote societal inequality. Story (1998) clarifies that correctly implemented "Universal Design" is often undetected since it integrates with the design process [19]. Being accessible to all user groups from the start of the design usability implies seamless implementation of Universal Design strategies [20]. Thus, suggesting a facility specifically for elderly/differently-abled negates the very concept of Universal Design.

Furthermore, Steinfeld and Danford (1999) establish that in spite of the origin of Universal Design since the mid-1970s, Universal Design concepts still lack practical implementation due to lack of adequate contextual theory [21]. Similarly, Steinfeld (1975) holds that the inclusion of empirical data gathered from human-centric research related to spatial behavior of differently-abled people is an ideal way to implement Universal Design solutions [22]. Thus, this paper includes human behavior and user experience as the base of research.

Architectural Planning, as described by Professor Dr. Suguru Mori from Hokkaido University is a discipline for "planning" architecture. It helps in conducting "practical problem interest (research approaching reality)" rather than "research problem interest (research for research)". Architectural Planning is probably the ideal pedagogy for research on Universal Design. In Architectural Planning, the three levels of design hierarchy are (a) site, (b) services, and (c) building. In Universal Design professional practice, the building level is better addressed by designers since the scale of the development is smaller in comparison to other factors. However, without inclusive sites and services, the individual buildings will be inaccessible to all; in spite of the individual building being universally designed [23]. Likewise, during the primary survey for this paper, even respondents prioritized site and services for implementing Universal Design considerations.

In relation to this, Universal Design considerations in an urban core with historic precincts are more challenging than any other urban scenario [24].

## 2.2. Fundamental Understanding of Accessibility

There are ideological differences in the way different nations perceive Universal Design, since Universal Design came into effect from 1985 (originating in the United States) and Barrier-Free Architecture has existed since 1974 (originating from Machida City, Japan). Akiyama and Kim (2005) establish that unlike the United States where Universal Design signifies disabled-friendly approaches, in countries such as Japan it is more holistic. Japan

uses Universal Design principles as a mode to facilitate the entire population. Relatively new initiatives such as "Transportation Accessibility Improvement Law, 2000" or old projects such as "Welfare model cities for the disabled, 1973" reflects Japan's focus on using Universal Mobility as an infrastructural as well as a social tool in enabling the environment for all [25]. In the similar lines, the authors also considered that the physical structure of an urban area directly impacts the walkability scenario [26].

However, in India, the situation is different from countries such as the United States or Japan. Likewise, Indian Universal Design Principles were published in 2011, which proposed five principles: (a) *Saman* (equitable), (b) *Sahaj* (usable), (c) *Sanskritik* (cultural), (d) *Sasta* (Economic), and (e) *Sundar* (aesthetic). The "Indian Principles" are different from the Seven Principles of Universal Design that were being followed in India until the inception of these principles [27]. Apart from political and administrative difficulties, a cultural stigma is also attached to the Universal Design thinking in India. Disability in India has been associated with past sins, and the disabled people are historically ignored in social/religious participation [28]. Thus, implementing Universal Design guidelines in India is substantially complex and requires an audit to assess the condition of inclusivity before imparting Universal Design. The audit shall specify the degree of accessibility required in a site-specific manner.

In addition, a modified performance assessment by creating a research agenda and involving professionals from different fields is the ideal way to practice Universal Design [29]. India and many developing nations have a general inception that the user group facilitated by Universal Design is considerably lesser than the ones "not facilitated" by its absence. However, the conditions that people experience while they are situationally disabled, such as a fracture, or pregnancies, or carrying a child, are seldom considered by these nations in urban-level infrastructural design. Another stigma related to Universal Design is the myth that it increases the project cost. Nevertheless, considering the global changing demographic pattern and inclusive planning considerations, the cost incurred due to Universal Design is utilitarian [30].

### 2.3. Universal Mobility in Urban Areas

Walkability conditions and transportation facilities are the two primary components of urban mobility. Nevertheless, poor infrastructure such as irregular footpath spaces and dissatisfactory pedestrian slope poses threat to elderly/differently-abled bodied and abled bodied alike. Additionally, any physical barriers in urban mobility are against the notion of "right to city or city for all" [31]. In the course of this research, the authors observed a similar phenomenon in central Kolkata. The photographs of a stretch in central (core) Kolkata showed in Figures 19–30 depict the dilapidated mobility conditions.

Along these lines, Frye (2014) argues that the global ageing population complemented by the falling birth rate is posing an infrastructural challenge to the increasing urban population. Frye's research identifies several factors that influence urban mobility: (a) overcrowding of vehicles/terminals, (b) uneven or broken road surfaces, (c) high kerbs/deep storm drains, (d) inaccessible public transport vehicles, (e) cost/affordability of public transport, (f) attitude of drivers and other staff, and (g) lack of accessible information. Frye validated these factors by correlating age and disability [32]. The authors used the factors from Frye's paper along with some contextual factors while conducting visual observation of the old core of Kolkata (elaborated in Section 3.3: Visual observation of an old city core). While transportation is widely researched, "walkability" for a diverse user group is a relatively restrictive topic in academia till date.

Similarly, Mori (2001) proposes that for researches involving a high human behavior interface, the research methodology should involve the feedback of the end-user from the beginning of the research [33]. Mori (2002) further explains that walkability involves an intricate relationship with space structure and mentions the needs to include children, elderly and disabled people in researches involving walkability [34]. Factors affecting user's behavior in urban street spaces have a positive or negative effect depending on

the genre of infrastructure and mobility condition of the user group [35]. Architectural planning research methods by comparing and/or relating the rational parameters (such as zoning and space allocation) with qualitative factors (such as pedestrian behavior), is an effective way to urban mobility-related issues in old core cities [36]. The space structures in old cities of developing nations such as India are complex due to their historic origin. These areas have high density, mixed land use, and a lack of space allocated to infrastructure.

Preiser (2008) argued that there is no comprehensive audit checklist for accessibility, and suggests that activities such as workshop and research group discussions for contextual accessibility assessments [37]. The authors took inspiration from Preiser's work and conducted eleven digital interactive sessions all across India to gain a contextual perspective on the topic. In addition to this, stakeholder specific approach in urban-level accessibility surveys by involving differently-abled people and elderly people enhances the effectiveness of an accessibility survey [38]. Furthermore, interviews held in Kolkata helped the authors understand the user's perspective about mobility in old Indian cities.

The learning elaborated in the Section 2.1 (Universal Design Theory), Section 2.2 (Fundamental Understanding of Disability), and Section 2.3 (Universal Mobility in Urban Area) are linked to the objectives, and the linkage is shown in Figure 2. Before starting the data collection and analysis, the authors for this paper created a knowledge base that helps to proceed further towards objectives, and subsequently answer the research question.

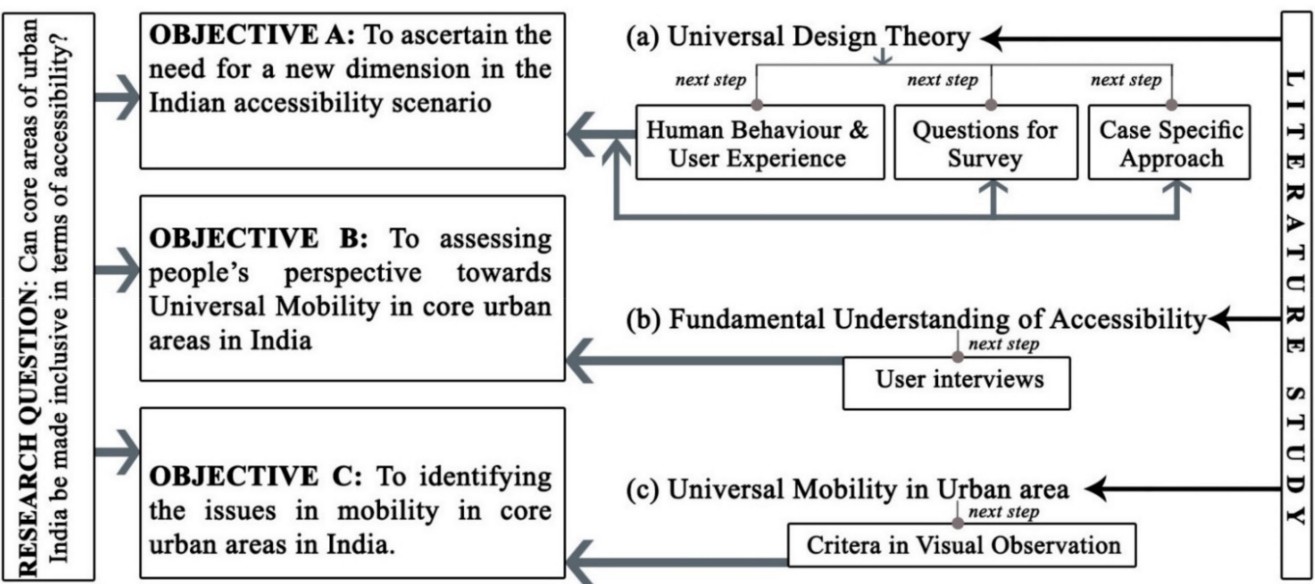

**Figure 2.** Linkage between research question and objectives with findings from the literature study (Source: Author).

## 3. Survey and Results

The details of the three surveys as mentioned in Section 2 are elaborated in this section of the paper. The authors have explained each survey in three basic parts: (a) survey process, (b) observation and analysis, and (c) findings and discussion.

### 3.1. Opinion of Design Fraternity in India

The design fraternity of a country, which includes architects, planners, designers, and civil engineers, are responsible for the task of nation-building in terms of infrastructure. Thus, the Indian design fraternity's opinion was essential for substantiating the aim of this research. The authors conducted eleven "virtual interactive platforms" (through ZOOM and Google Meet platforms) in various parts of India for discussing the intent of this research with the design fraternity of India. The platforms included workshops, seminars, studio, Technical Education Quality Improvement Program (TEQIP), and lectures. Table 1 shows the details of the eleven virtual interactive sessions.

**Table 1.** List of interactive sessions in India between June and November 2020, for substantiating the aim of the research (Source: Author).

| S. no. | Date | Type of Session | Title of Session | Location |
|---|---|---|---|---|
| 1 | 25–28 June | Summer Studio | Universal Design | SAKHA, Nagpur |
| 2 | 27–29 July | Workshop | Being Creatively Rational | School of Architecture and Interior Design, SRM Institute of Science and Technology, Kattankulathur |
| 3 | 07–09 August | Technical Education Quality Improvement Programme | Universal Design: Architecture for All | Department of Architecture, Madhav Institute of Technology and Science, Gwalior |
| 4 | 14 August | Webinar Series | Universal Design and Accessibility—A route towards Sustainability | Jawaharlal Nehru Architecture and Fine Arts University, Hyderabad |
| 5 | 19–21 August | Workshop | An Architect's Approach towards Universal Design and Accessibility | School of Architecture, DY Patil University, Pune |
| 6 | 28 August | Semester Coursework Expert Lecture | Universal Design Approach | Akhil Bharatiya Maratha Sikshan Parishad's Anantarao Pawar College of Architecture, Pune |
| 7 | 05 October | Webinar | Universal Design | School of Architecture, Central University, Ajmer |
| 8 | 06 October | Guest Lecture | Being Creatively Rational | Rajalakshmi School of Architecture, Mevalurkuppam |
| 9 | 06–08 October | Semester Coursework Expert Lecture | Universal Design and Accessibility | Marathwada Mitramandal's Institute of Environment and Design's College of Architecture, Pune |
| 10 | 28–29 October | Workshop | Being Creatively Rational | School of Architecture, Delhi Technical Campus, Noida |
| 11 | 08 November | Institute of Town Planning Annual Lecture | Assessing accessibility for equitable planningwith focus on disabled and elderly | Institute of Town Planners India, Kolkata |

Architects, planners, designers, government officials, architecture students, and planning students, attended these virtual interactive platforms. Although there were eleven venues, the participants came from over sixty cities across India. The major content of the workshop included (a) introduction to Universal Design and its salient features, (b) the International and National Guidelines on Universal Design, (c) anthropometrics and ergonomics in Universal Design, (d) the application of Universal Design in different building types and streetscape, and (e) accessibility audit. The authors were the primary mentors for the sessions, occasionally complemented by other experts from the field of accessibility and spatial design. The major findings from this workshop included (a) consciousness about the dissimilarity between Universal Design, Barrier-Free Standards, and Inclusive Design, (b) facilitating participants towards using the "Universal Design" principles in Architectural Design and Urban Planning, and (c) realizing that in architecture and planning, Universal Design is not a choice, but a prerequisite. The authors included interactive exercises and idea exchange sessions in the lecture sessions. At the end of each session, the participants acquired ideas about applying Universal Design in Architectural Design/Planning. The participants also received the basic idea of Accessibility Audit as a predecessor for Architectural Design. Figures 3 and 4 show some glimpses from the "virtual interactive platforms".

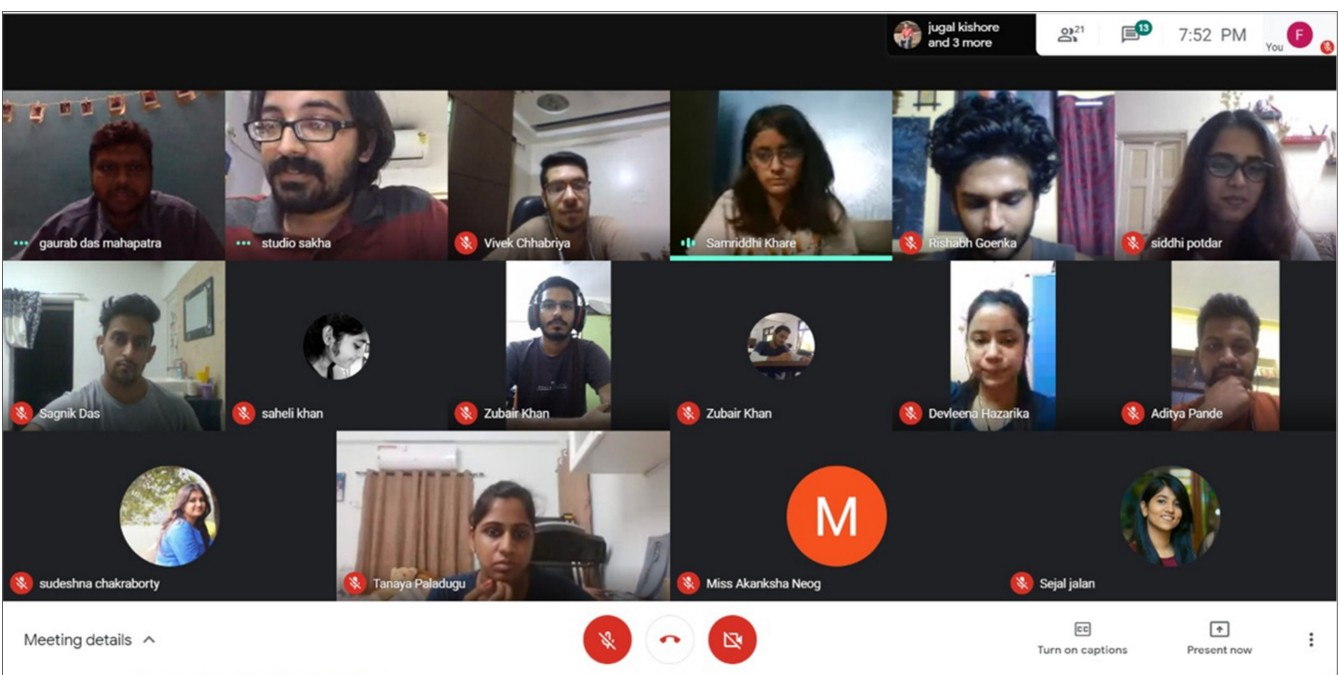

**Figure 3.** Screenshot of session on day 2 from the Summer Studio by Studio Sakha, Nagpur, India from 25–28 June 2020 (Author: Top left) (Source: Author).

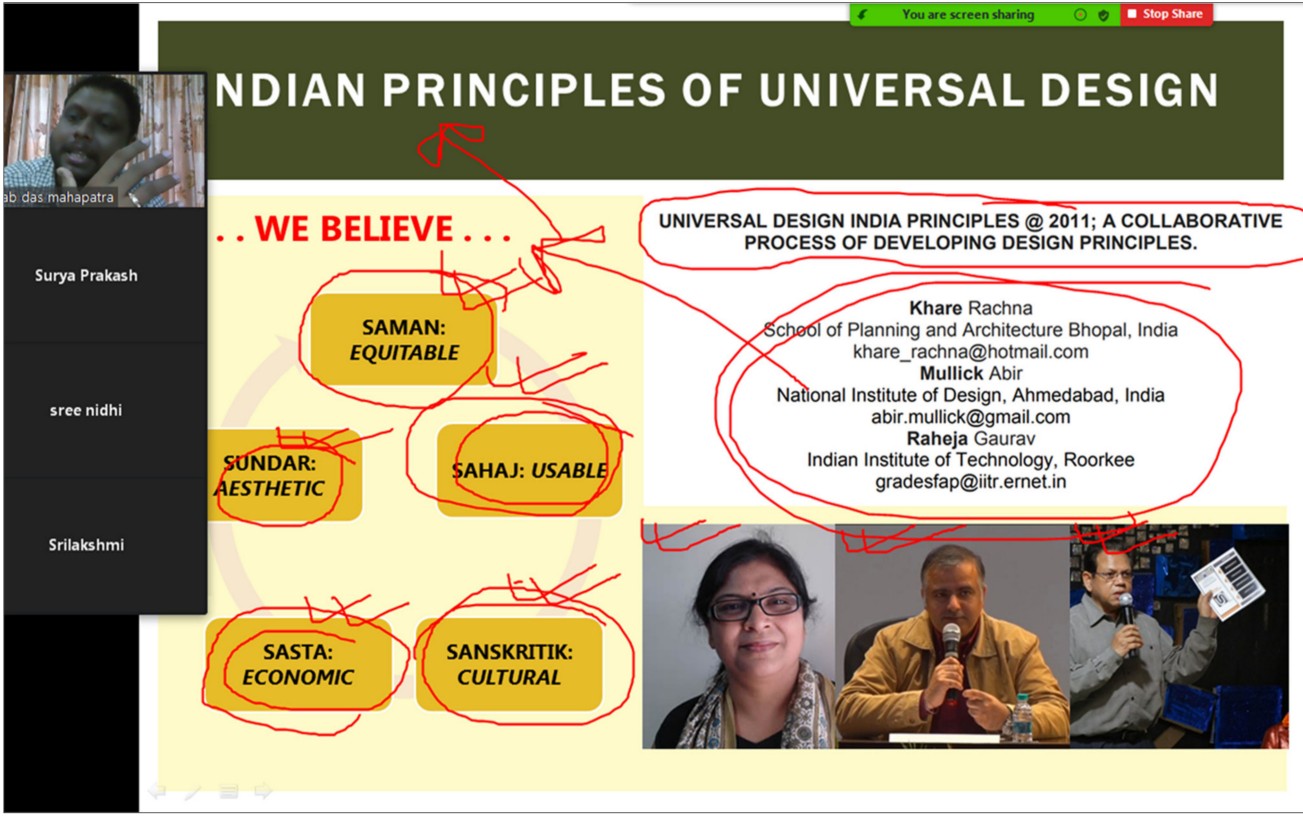

**Figure 4.** Screenshot from the workshop by Rajalakshmi School of Architecture, Mevalurkuppam, India on 6 October 2020 (Author: Top left) (Source: Author).

### 3.1.1. Observation and Analysis

To further analyze the aforementioned points related to workshop learning, the authors created a "Google Form" questionnaire (https://forms.gle/Cx5v4VNsiTV8LQQ28 (accessed on 1 March 2021) for the participants at the end of each session. The questionnaire inquired about certain aspects of Universal Design: (a) position of Universal Design in the design domain, (b) the prioritization of site and services in Universal Design, (c) rating national policies, (d) the level of difficulty in a Universal Design scenario in old cities, (e) the position of "Transportation" and "Accessible Information" in the Universal Design domain, (f) the difference in the Universal Design scenario between old and new cities, and (g) the need for a "Rating System". 14 questions ranging from multiple choice questions, dichotomous questions and Likert scale typology, were used in the questionnaire. The total number of respondents in this Google Form questionnaire was 310.

A total of 79% of the respondents prioritized Universal Design, and not Barrier-Free Architecture or Inclusive Design, in the field of Architecture and Planning [Refer to Figure 5].

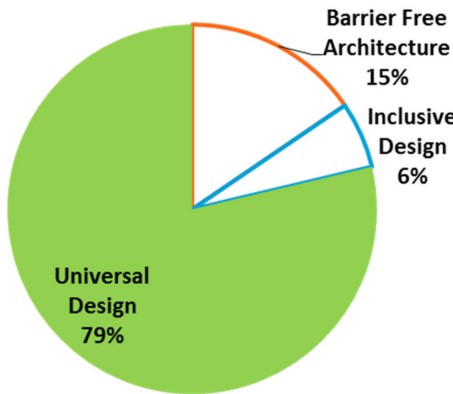

**Figure 5.** Most significant aspects in Architecture and Planning (Source: Author).

The respondents' priority in Universal Design in the Indian context is services (prioritized by 46%), followed by site (prioritized by 42%) [Refer to Figure 6]. A significant number (63%) of respondents advocated for different Universal Design guidelines/principles/audit formats, in old and new cities [Refer to Figure 7].

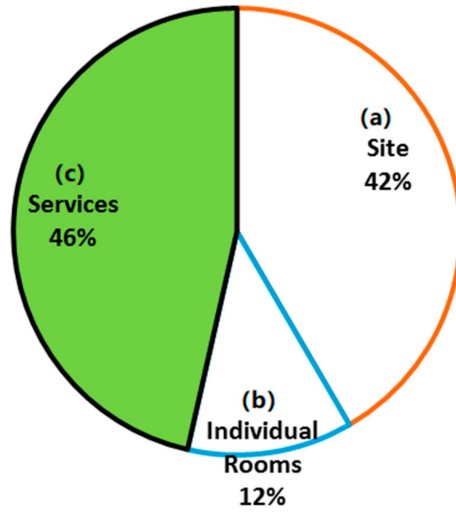

**Figure 6.** Preferred first rank of importance for the components in Universal Design in the Indian context: (**a**) site, (**b**) individual rooms (individual interior spaces), and (**c**) services (Source: Author).

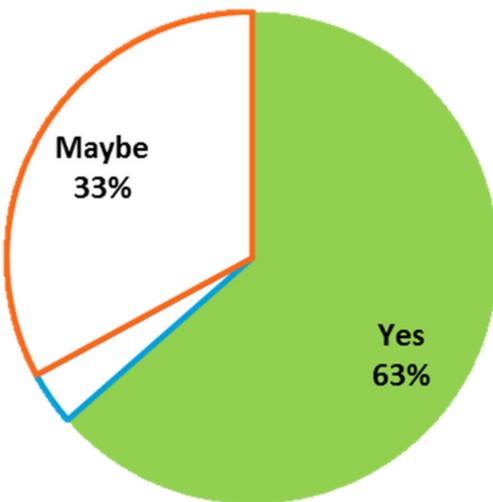

**Figure 7.** Do we need different Universal Design guidelines/principles/audit formats for old cities and new cities in India? (Source: Author).

The majority of the respondents stated that accessibility and Universal Design scenario in old core cities of India are difficult to impart [Refer to Figure 8]. Other than 8.07% of the respondents, all others specified a difficulty level of 5 or more on a Likert scale of 1–10 (where "1" is least difficult and "10" is most difficult).

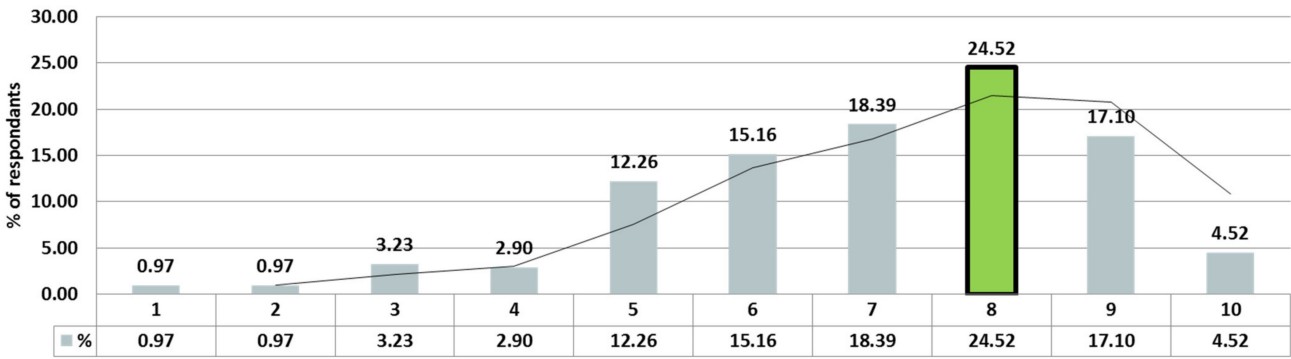

**Figure 8.** How difficult is it to impart the accessibility and Universal Design scenario in old core cities of India? (1—least difficult; 10—most difficult) (Source: Author).

The majority of the respondents also affirmed the Accessible India Campaign as of satisfactory status [Refer to Figure 9]. A total of 7.42% of respondents scored less than five on a Likert scale of one to ten (where one is least marked and ten is highest).

Out of the three components of the Accessible India Campaign, (i.e., built environment accessibility, transportation system accessibility, and information and communication eco-system accessibility), the respondents prioritized transportation system accessibility. Thus, transportation requires accessible and Universal Design features more than other segments of urban life. During this survey, 97.10% of the respondents also stated that like "Green Rating" (like LEED of USA, or GRIHA of India, or CASBEE of Japan) in Sustainable Architecture, a customized rating system for Universal Design is required as well.

### 3.1.2. Findings and Discussion

The pan-India data collected from the survey through Google Form questionnaire explicated certain aspects of Universal Design in the Indian context. The involvement of the Indian design fraternity made the study more fruitful, since these are the people who

spread the awareness of Universal Design. Table 2 shows a summary of the findings from this exercise.

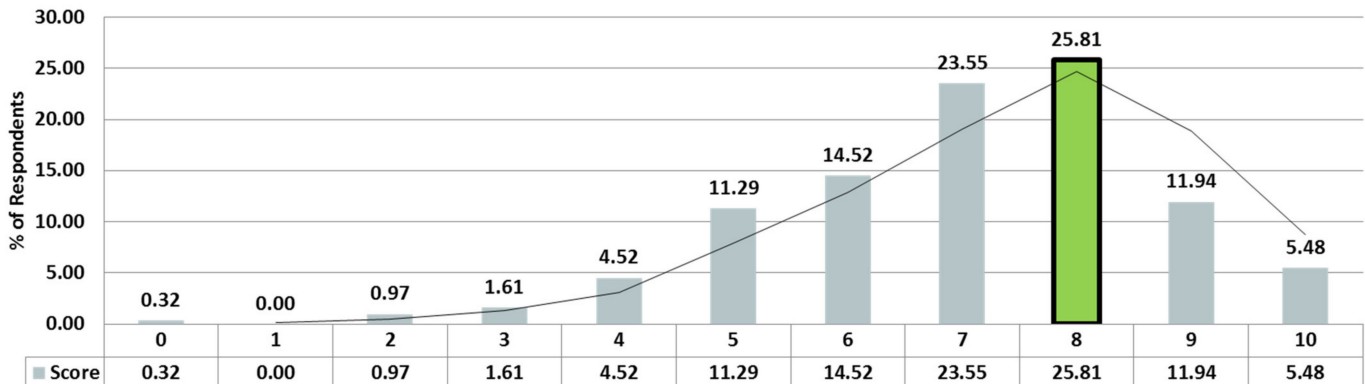

**Figure 9.** How much do you rate "Accessible India Campaign" in terms of Universal Design and Accessibility (0—least; 10—highest) (Source: Author).

**Table 2.** Learning from the interactive sessions in India between June and November 2020 for substantiating the aim of the research (Source: Author).

| S. No | Topic | Learning |
|---|---|---|
| 1 | Importance of Universal Design in Architecture and Planning | Unlike a few decades back, Barrier-Free architecture and Inclusive Design are relatively outdated. The present need for India is Universal Design. |
| 2 | Prioritization of site level and services in Universal Design | Universal Design is necessary, and within it, "SITE and SERVICES" should be the priority. However, the focus of all existing Indian guidelines is "Building Specific" or rather, towards Individual enclosed spaces. |
| 3 | Differences in an accessibility scenario between old core cities and new planned cities in India | In India, new cities and old cities need different Universal Design guidelines, calling for constructive criticism of the latest national policies. |
| 4 | Level of difficulty in old core cities in India | Different Universal Design guidelines are required for old and new cities due to their differences in spatial evolution/temporal growth, institutional mechanisms, infrastructural patterns, and demographic reasons. |
| 5 | Impact of present national campaigns towards inclusive planning | Although National Policies are satisfactory, initiating a case-specific approach or a flexible assessment pattern is ideal. |
| 6 | Position of "Transportation and Accessible Information" in Universal Design | Pan-India, accessible transportation is an issue. Instead of focusing on building or technology-oriented accessibility, the government should focus on making the transportation system accessible. In infrastructural terms, accessible streets and accessible mobility is needed. |
| 7 | Need for customized rating system in Universal Design | A new "Rating/Indexing" system in Universal Design makes the Universal Design scenario more quantitative in analytical terms. Thus, the scope for intervention in infrastructural terms based on the data derived from rating systems might be used for promoting inclusiveness. |

Each of the findings as mentioned in Table 2 shall form the basis of further investigation for respective focus area.

### 3.2. People's Perception in an Old City Regarding Walkability and Mobility

After gathering the Indian design fraternity's opinion, the authors conducted a survey for accessing people's perception about walkability and mobility, involving people from Kolkata. Kolkata, besides being a British colonial city, is also nearly 350 years old [39].

Thus, Kolkata serves as an ideal case for studying people's perception of mobility in an Indian old city. Due to coronavirus pandemic, India experienced a nationwide lockdown (complete and partial) from 22 March 2020 until 30 November 2020; this meant restrictive

movement for the authors. Thus, the authors collected samples over a period of five months from 17 July 2020 until 2 November 2020, via online mode besides face-to-face mode.

The intention of this survey was to recognize (a) the frequency and purpose of outdoor mobility, (b) people's perception of a public transportation system, (c) the status of walkability, and (d) awareness of national campaigns.

### 3.2.1. Data Analysis

Considering the aforementioned intentions, the authors prepared a Google Form questionnaire (https://forms.gle/oB9MvhWSfQ6PZAKk9 (accessed on 1 March 2021)). Like the previous Google Form in the case of "Opinion of Design Fraternity in India", the questions in this survey also ranged from Multiple Choice Questions, Dichotomous Questions, and Likert Scale based. The total number of respondents for this survey was 125, which were acquired from 40 wards within KMC limits.

Mr. Deoraj Pande (resident of Ward 44, aged 74) who goes out for his house daily work in spite of his severe arthritis prefers using private transport for commuting. Mr. Barid Baran Mahaty (resident of Ward 48, aged 58), a "Person with Disability" (PwD) cardholder, prefers using local trains over any other means of public transport. Mrs. Kanaklata Chakraborty (resident of Ward 50, aged 76), another PwD cardholder, expressed her grief regarding the difficulty in identifying address which becomes aggravated due to her partial blindness. Mrs. Dugarani Chaudhury (resident of Ward 45, aged 96) has been under restricted movement for the last two decades and avoids public transport in case of occasional outings. Mr. Prasanta Das (resident of Ward 48, aged 61), who received a PwD card after becoming mobility impaired, mentioned about the dilapidated condition of the streets. Difficulty in the waiting facility in public transportation has been identified as an alarming situation by Mrs. Manisha Roy (resident of Ward 43, aged 72), Mr. Ram Adhikari (resident of Ward 49; aged 67), and many other respondents.

After taking such opinions from respondents, the authors filled the Google Form for the respondents, in their presence. for better communication with locals, the authors involved one local architecture student (named Ms. Disha Maity) and a retired municipal worker from Kolkata(named Mrs. Manorama Mahanty) in their survey. Respondents became more vocal about their issues and problems when communicated in the local language by local people, than during formal Google-form based interview. A bilingual survey format was also used for communicating with locals and their comments were noted in the local language "Bengali" [shown in Figure 10], which was later translated into English, at the "Laboratory of Architectural Planning" at Hokkaido University.

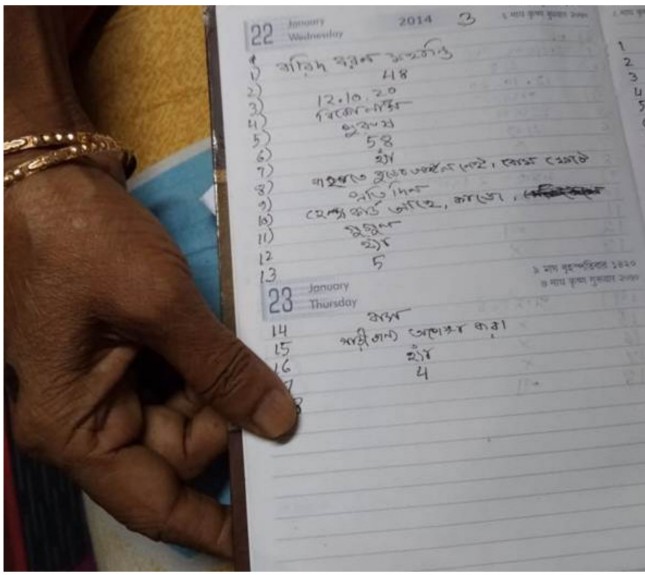

**Figure 10.** Notes were taken in Bengali and later translated into English (Source: Author).

The last survey titled "Opinion of Design Fraternity in India" indicated Universal Design as the need of Indian design scenario. Universal Design encompasses the need for a diverse group. Thus, in spite of focusing on elderly and differently-abled people, the authors collected samples from all categories.

Out of the 125 respondents, 48% of the respondents were male, 50.40% were female, and one respondent belonged to the third gender. However, in terms of age groups, the focus was on the senior citizen [Refer to Figure 11]. A total of 33.60% of the respondents belonged to the age group of sixty years or above sixty years, 28.80% from the age group between twenty and thirty-five years, 20.80% from the age group between fifty and sixty years, 16% from the age group between thirty-five and fifty years, and 0.80% from the age group of ten years or below ten years.

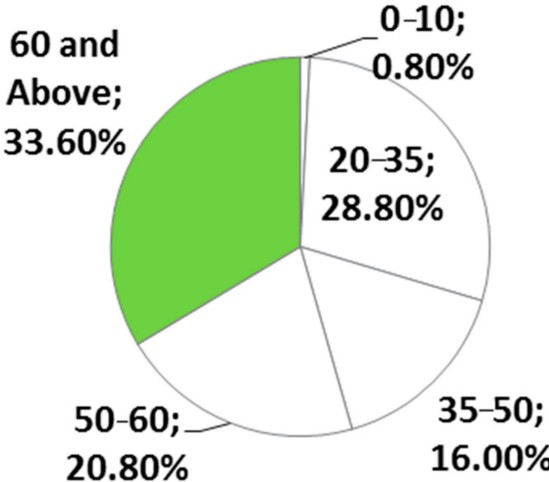

**Figure 11.** Composition of the respondents in terms of age (Source: Author).

Out of the 125 respondents, 83.20% were able-bodied and 16.80% were differently-abled. Out of the 16.80% differently-abled people, only 3.2% of them possessed a "Person with Disability card". "Person with Disability card" is also known as PH/disability/handicap certificate, and this certificate is issued by competent medical authority (in India) specifying the type and extent/severity of the cardholder's identity. As of January 2021, "Person with Disability card" is being replaced by UDID (Unique Disability Identity) with a view of creating a national database for persons with disabilities. 3.2% of the surveyed respondents were "medically" recognized "disabled" or differently-abled, which is above the national percentage of 2.21%.

The types of medical issues that the respondents were facing are (a) low vision, (b) locomotor disability, (c) arthritis, (d) asthma and (e) terminal illness (cancer). Table 3 elaborates the type of disability and recognition status as per "Right to Persons with Disabilities Act, 2016".

**Table 3.** Types of respondents' disabilities (Source: Author).

| S.No | Type of Disability | Recognized by "Right to Persons with Disabilities Act, 2016" | Number of Respondents | Percentage of Respondents |
|------|--------------------|------------------------------------------------------------|-----------------------|---------------------------|
| 1 | Low Vision | Yes | 6 | 28.57 |
| 2 | Locomotor Disability | Yes | 11 | 52.38 |
| 3 | Arthritis | No | 2 | 9.52 |
| 4 | Asthma | No | 1 | 4.76 |
| 5 | Cancer | No | 1 | 4.76 |

Hereafter, "senior citizen and differently-abled people" are paired in a single group and compared with "able-bodied people under the age of sixty". The "able-bodied people under the age of sixty" are referred to as "Category A" and "senior citizen and differently-abled people" are referred to as "Category B". The total number of "able-bodied people under the age of sixty" was 74, which is 59.2% of the total respondents. The total number of "senior citizen and differently-abled people" was 51, which is 40.8% of the respondents. For the rest of the discussion about "People's Perception in an old city about walkability and mobility", the "Able-bodied people under the age of sixty" and "senior citizen and differently-abled people" are referred to as "Category A" and "Category B" respectively. The various segments of the questionnaire are discussed hereafter:

- Frequency and purpose of going out.

First, the authors asked the respondents about the respondents' frequency of going out of their house. Only 50.98% of "Category B" ventured out of their houses on a daily basis, in comparison to 75.68% of "Category A". Likewise, the "weekly", "monthly", and "yearly" rate of going out was more in "Category B" as compared to "Category A". Figure 12 shows the options selected by both "Category A" and "Category B" regarding their frequency of going out of respective houses.

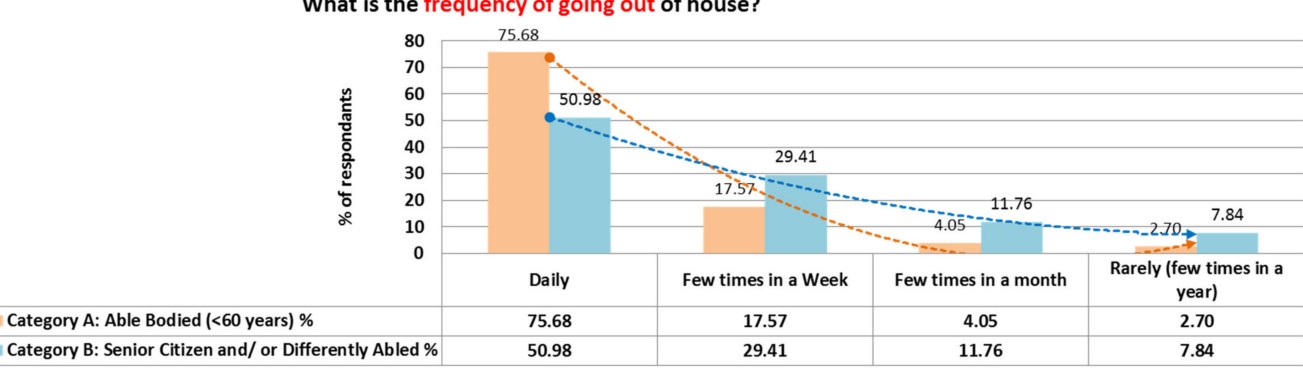

**What is the frequency of going out of house?**

| | Daily | Few times in a Week | Few times in a month | Rarely (few times in a year) |
|---|---|---|---|---|
| Category A: Able Bodied (<60 years) % | 75.68 | 17.57 | 4.05 | 2.70 |
| Category B: Senior Citizen and/ or Differently Abled % | 50.98 | 29.41 | 11.76 | 7.84 |

**Figure 12.** Frequency pattern of going out of respective house among different user groups (Source: Author).

After determining their frequency pattern of going out of their house, the authors questioned respondents about their purpose of going out. The options provided were (a) professional/academic, (b) medical reasons, (c) social gathering, (d) availing government benefits (such as health cards), and (e) daily household work. The respondents could choose all the options that were applicable. Figure 13 shows the options selected by both "Category A" and "Category B" in relation to their purpose of going out of respective houses. Both "Category A" (89.19% of them) and "Category B" (62.75% of them), mentioned "social gathering" as the most common reason for going out of the house. Respondents from "Category A" showed a higher share (75.68%) compared to "Category B" (47.06%), in going out of the house for professional or academic purposes. "Category B" showed a higher share (50.98%) compared to "Category A" (13.51%) in medical-related outings. No respondent from "Category A" went out of the house for availing Government benefits (such as health cards). A remarkable 56.86% of "Category B" went out of the house for daily household works, in comparison to 60.81% of "Category A".

- Searching/identification of an address.

Next, the authors discussed street-level cognition with the respondents. The question was how the respondents usually search/identify an address. The options provided were (a) using online services such as Google maps, (b) asking people on the streets, (c) identifying landmark (such as the nearest junction or, old tree or, famous temple), and (d) referring to street signage. Although people in urban areas frequently use all the aforesaid options, the authors asked respondents to mention their most preferred

option. Figure 14 shows the options selected by both "Category A" and "Category B" in relation to their preferred options while searching/identifying an address. The majority of respondents from "Category A" (56.76%) preferred "online services" to identify/search an address. In response, the majority of respondents from "Category B" (45.10%) choose "asking people on the streets". Respondents from both the categories showed noticeably less dependence on referring street signage, with 1.35% of respondents from Category A and 3.92% of respondents from Category B. More respondents from "Category B" (25.49%) preferred "Identifying landmark (such as the nearest junction or, old tree or, famous temple)" as their means to identify an address, in comparison to only 9.46% of respondents from "Category A".

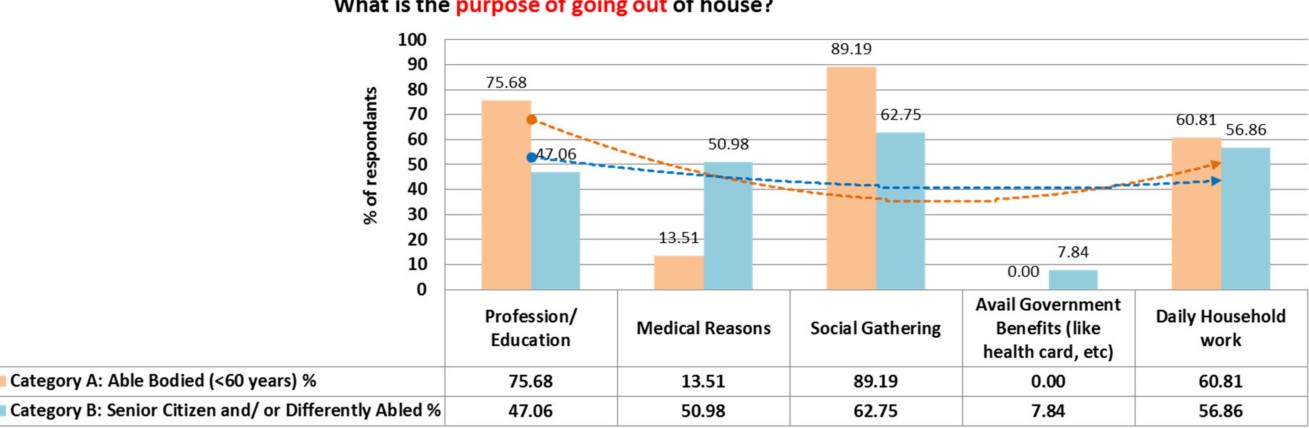

**Figure 13.** Purpose of going out of the house among different user groups (Source: Author).

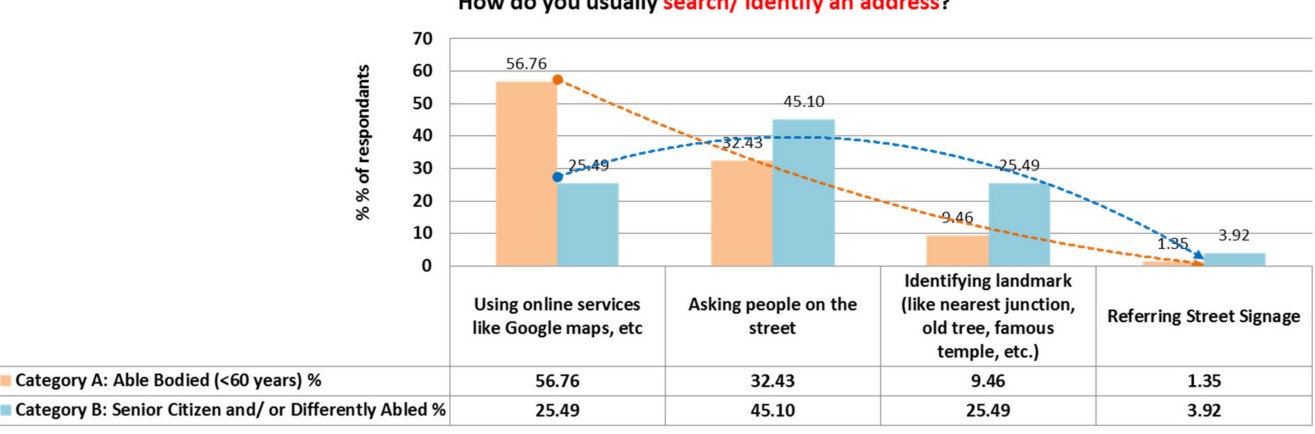

**Figure 14.** Preference in the identification of an address (Source: Author).

- Public Transportation.

The perception of people in public transportation was the next topic of investigation. For this purpose, respondents from both "Category A" and "Category B" were primarily asked if they use public transport. If the respondents did not use public transport on a daily basis, frequent or periodic usages were also taken into consideration for data input. A total of 54.90% of the respondents from "Category B" used public transport, in contrast to 78.38% of respondents from "Category A". The respondents from both categories who do not use public transports are either having their own mode of transport (such as four-wheeler or two-wheeler), or having designated vehicle from their place of work, or are dependent on rental cab services, for example, Uber, Ola, and Rapido).

Thus, for further study on the topic of public transportation, the authors considered only 78.38% from "Category A", i.e., 58 respondents, and 54.90% from "Category B", i.e., 28 respondents.

The authors asked these respondents about their comfort level in using public transportation. A Likert scale approach was undertaken for this purpose and respondents rated their comfort level on a scale of one to ten. Score one signified lowest comfort and score ten signified most comfort. The weighted mean score is 5.39 for "Category B" in comparison to 6.17 for "Category A".

The next question was about modal preference. The authors asked these respondents for specifying which mode of public transport in their city the respondents often used. The options provided were (a) bus, (b) tram, (c) metro rail, (d) local train, (e) auto rickshaw, and (f) cycle/hand-pulled rickshaw. Kolkata having a multiplicity of transport mode and a well-connected transportation network encourages its residents for using multiple modes in a single origin-destination route. However, the authors asked respondents to mention the mode which they used most. Figure 15 shows the options selected by both "Category A" and "Category B" in relation to their most preferred public transportation mode. Respondents from both "Category A" (48.28% of respondents) and "Category B" (67.86% of respondents) mentioned Bus as their most preferred mode of public transportation. Tramway, often referred to as the heritage of Kolkata, was not given any preference by either category of respondents. Preference of Metro Rail was comparatively similar in "Category A" with 18.97% of the respondents and "Category B" with 14.29%. However, 10.71% of respondents from "Category B" preferred local train, in comparison to only 5.17% of respondents from "Category A". A total of 24.14% of respondents from "Category A" and 7.14% of respondents from "Category B" preferred auto-rickshaw, respectively. Although 3.45% of respondents from "Category A" choose Cycle/Hand-pulled rickshaw as their preferred mode of public transport, none of the respondents from "Category B" opted for this.

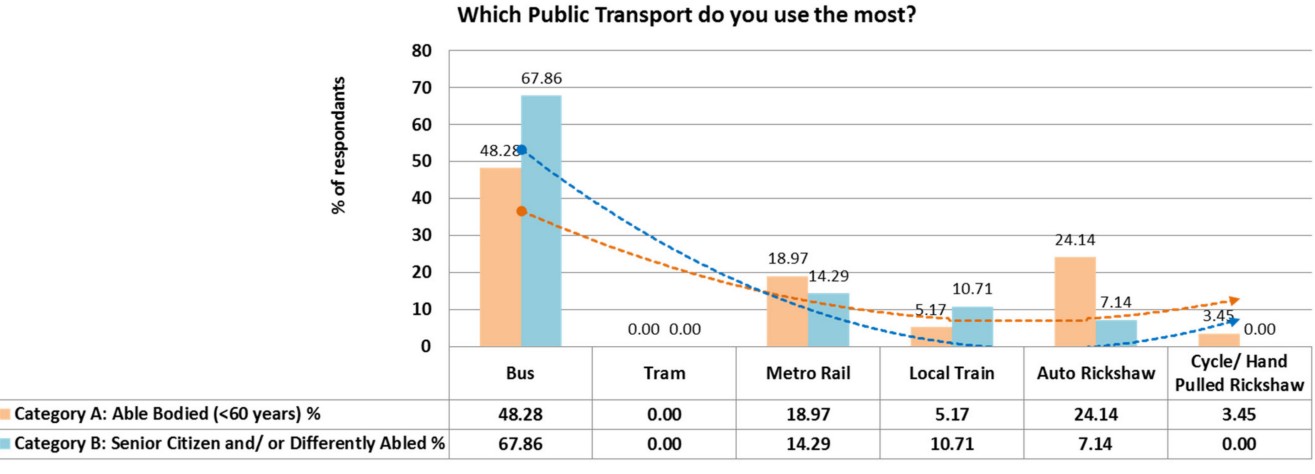

**Figure 15.** Preference in the usage of transportation mode (Source: Author).

The last question in the category of public transportation was identifying the problem in accessing public transportation, by individual respondents. For this question, the authors asked the respondents for choosing any one option from the following, as their major problem in accessing public transportation: (a) difficulty in the waiting facility (a poorly designed bus stop or, improper visual notification), (b) reaching public transport from home, (c) any other (such as congestion or, misbehavior by co-passengers), and (d) fare. Figure 16 shows the options selected by both "Category A" and "Category B" in relation to their major problem in accessing public transport. The majority of respondents from both "Category A" (62.07%) and "Category B" (46.43%), stated "Difficulty in the waiting facility (such as a poorly designed bus stop or, improper visual notification)," as their major

problem in accessing public transport. The second preference for respondents from both categories (Category A: 18.97%; Category B: 39.29%) was the issue of "reaching the public transportation from individual homes". Relatively similar share both the categories, 13.79% respondents from Category A and 14.29% from Category B, selected "Any other (such as congestion or, misbehavior by co-passengers)". Although 5.17% of the respondents from Category A mentioned "Fare" as their reluctance towards accessing public transport, no respondents from Category B choose this option.

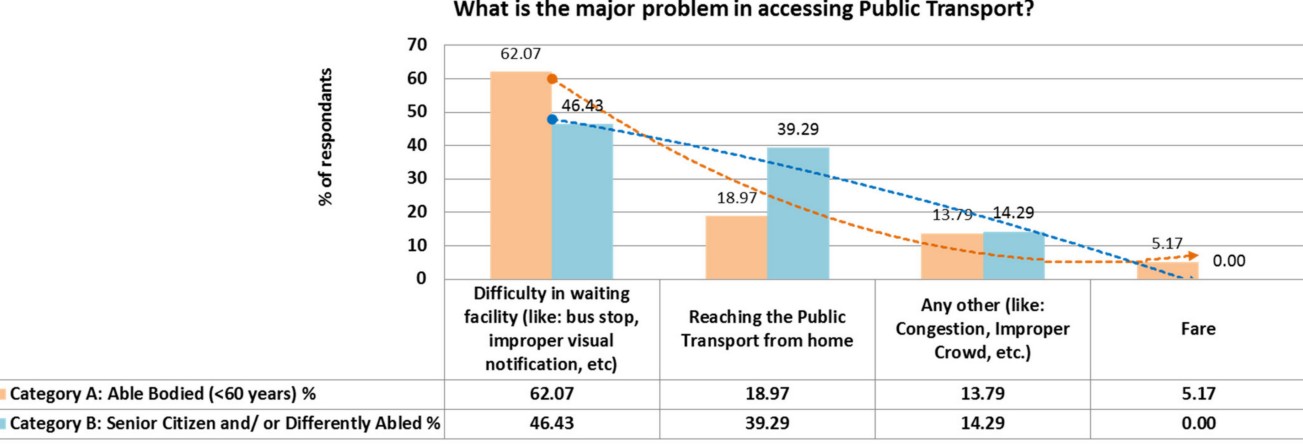

**Figure 16.** Difficulty in accessing public transportation (Source: Author).

- Walkability

    After acquiring ideas about public transportation, the authors investigated "walkability" from all 125 respondents (74 from Category A and 51 from Category B). Only 74.51% of the respondents from Category B walk/access the streets, in comparison to 93.24% of able-bodied people. The people who do not walk are the ones with a constant source of a private vehicle, having medical issues, or are dependent on others for their transportation. For the next question, the authors considered only that 93.24% from "Category A", i.e., 69 respondents, and 74.51% from "Category B", i.e., 38 respondents.

    The next inquiry in the walkability segment was related to the respondents' comfort level during walking on the streets. Using a Likert scale approach, the respondents rated their walkability experience on a scale of one to ten. Score one signifying lowest comfort and score ten signifying most comfort. The weighted mean score is 4.92 for "Category B" and 5.0 for "Category A".

- Awareness about National Schemes.

    The authors asked all the 125 respondents about their awareness about the existing national policies/initiatives related to accessibility and Universal Design, the latest of which being "Accessible India Campaign" or "Sugamya Bharat Abhiyan". A total of 92.16% of the respondents from Category B were not aware about the "Accessible India Campaign", in comparison to 85.14% of able-bodied people.

### 3.2.2. Findings and Discussion

The data collected by interviewing 125 people from forty wards of KMC helped in assessing people's perspective in the domain of Universal Mobility. The term "Universal Mobility" was not used during the interview; however, the authors posed the questions in a strategic pattern for assessing the scope of Universal Mobility. The four sets of inquiries as discussed in Section 3.2.1 portray an overall negative picture in the domain of mobility. Table 4 shows a summary of findings from the responses recorded by the authors during this survey.

**Table 4.** Learning from the interview sessions in Kolkata, India between July and November 2020 for substantiating the aim of the research (Source: Author).

| S.No | Topic | Learning |
|---|---|---|
| 1 | Frequency and purpose of going out | <ul><li>Health-related issues outside the preview of "registered" disabilities need recognition for availing health benefits. In any case, people with disabilities/senior citizen venture out daily. The rates of going out daily in able-bodied are more than those of their counterparts.</li><li>Senior citizens/differently-abled people have a higher rate of social gatherings than other purposes, hinting towards a need of promoting "sociable streets and public environment" which is in coherence with the United Nations Sustainable Development Goals.</li><li>Since old cities usually contain all the types of building use (as per occupancy), mobility corridors require equal or equivalent importance.</li></ul> |
| 2 | Searching/ identification of an address | <ul><li>The affinity of senior citizens/differently-abled people to rely on "asking people on the streets", is hinting towards the need for inclusive streets which not only enables but empower them as well.</li><li>The need for improving street signage (visual and auditory).</li><li>The domain of "online search" in terms of Google Maps can be improved by adding "accessibility" layer prepared by Architectural Planning only.</li><li>In spite of being developed on the British planning principle, the lack of "landmarks" shows the complexity in the urbanscapes, which can be addressed by introducing street elements as new icons of the urbanscape.</li></ul> |
| 3 | Walkability | <ul><li>Dissatisfaction with the use of public transport is due to (1) poor provision from state/central governments in the field of urban transportation and/or (2) better services from private players, for example, Uber, Ola, and Rapido.</li><li>The major transport mode is "Bus" and multiple locations are present for the bus to halt briefly. However, "bus stop" or "bus shelter" was scantily present. Clearly, the survey explains the need for prioritizing bus-related accessible infrastructure before developing any other modes.</li><li>The proposal for developing "Urban Accessibility Facility" can be prioritized in the topic of improving the "waiting facility" for bus.</li><li>The target for architects and planners should be facilitating the waiting time and emergency requirements especially for elderly and differently-abled.</li><li>The reason for the underutilization of Para-transits can be dealt with in a separate survey.</li></ul> |
| 4 | Awareness about National Schemes | <ul><li>A large percentage of walkability in spite of dissatisfactory pedestrian facilities proves that in developing countries, infrastructure is less considerable than daily need.</li><li>When people commute in spite of poor infrastructure, the marginalized users are compromised.</li><li>Relating to the fact that people are unaware of national policies and unknown about the scopes of better design, the motto of this research is "To create a market for the design, rather than designing for the market".</li></ul> |

### 3.3. Visual Observation of an Old City Core

Taking indications from the survey involving 125 people from Kolkata, a stretch located within the core of Kolkata was selected by the authors for further investigation. The focus of this exercise was assessing the Universal Mobility conditions in the old core of Indian cities. A visual survey for the selected stretch was conducted in the month of September and November 2020. The selected stretch is a part of Bipin Behari Ganguly Street (Google Map link: https://goo.gl/maps/ph8wWJWiCzY1TgEH8 (accessed on 1 March 2021). In the Master Plan of Kolkata, the land use for this area is demarcated as "Mixed Use" [40]. It is to be noted that mixed land use refers to the co-existence of more than one land use on a single stretch; for example, residential and industrial buildings in

a single street. Mixed-use also refers to the presence of multiple "single buildings" with different building use on different floors.

The selected stretch is observed in multiple old cartographic evidences of Kolkata since 1785 CE, as shown in Figure 17.

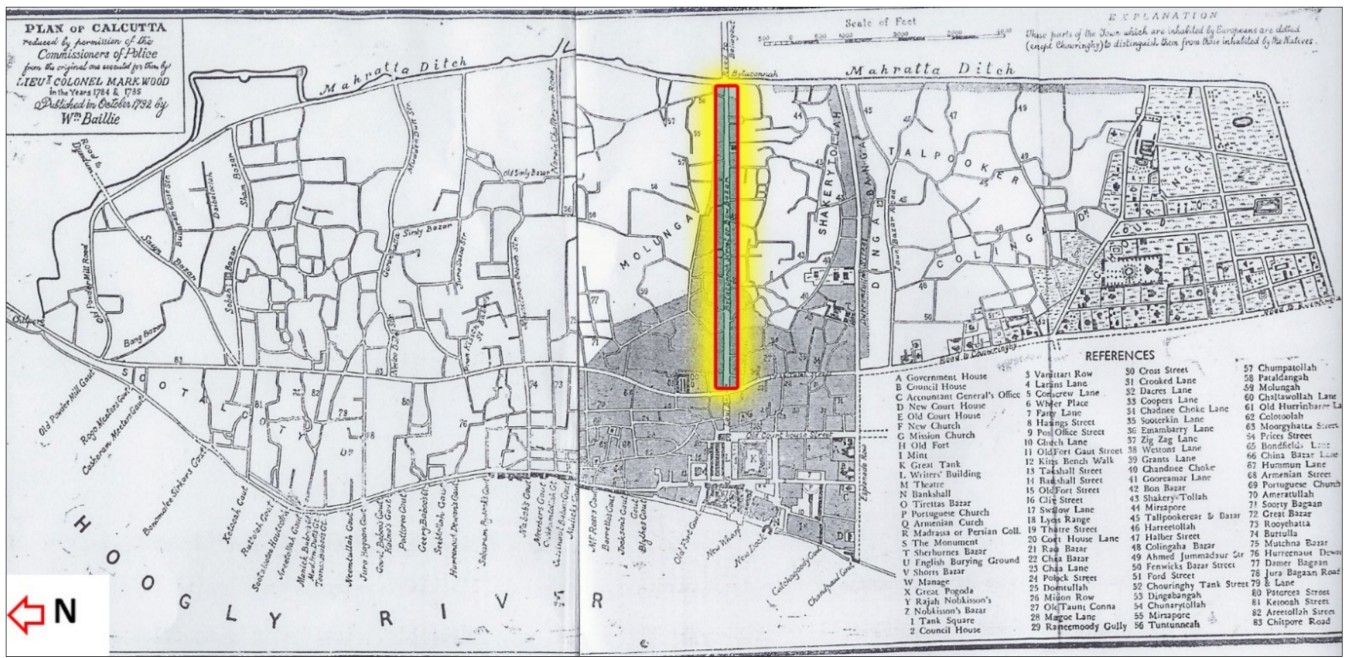

**Figure 17.** Map of Kolkata (Calcutta) by Lt. Col. Mark Wood in 1785; the study stretch selected for the visual survey is demarcated is highlighted. (Source: Harvard Library Online).

### 3.3.1. Visual Survey

The authors selected a part of the 300-year-old stretch (shown in Figure 17) for the visual survey. This stretch has undergone minimum temporal change due to its historic nature. The stretch is further shown in Figure 18, where "A" and "B" are starting and ending point of the study stretch, respectively. Point A is Bowbazar Crossing and Point B is Bentinck Street Crossing.

In the policy document for "Accessible India Campaign", the component of "Transportation System Accessibility" has the objectives of enhancing proportions of (a) accessible airports, (b) railway stations, and (c) public transport. The part of Kolkata, which is being proposed for the Research, does not have an airport, however, there is a Railway Station and multiple Underground Metro Rail Stations. Further, there are multiple modes of other transport available here, including (a) tramways, (b) bus, (c) auto, (d) hand-pulled rickshaw, and (c) taxi/cab. In the city land-use master plan, this area of Kolkata is demarcated as mixed land use zone. The stretch predominantly consists of buildings with multiple uses, mostly residential buildings clubbed with business/commercial establishments. However, buildings with institutional, educational, assembly, mercantile, and storage uses were also present. Moreover, a substantial number of buildings in this stretch were heritage buildings [refer to Figure 19].

Multiplicity in building use causes heavy pedestrian footfall throughout the day in this stretch. Thus, this stretch serves as a suitable location for a visual survey for the assessment of mobility conditions in an old Indian city. A social worker (named Mr. Akash Das) from Kolkata assisted the authors in identifying issues from a local perspective. The photographs taken during the visual survey were later interpreted at the Laboratory of Architectural Planning, Hokkaido University. Figures 20–30 illustrates how the stretch is unfavorable for Universal Mobility.

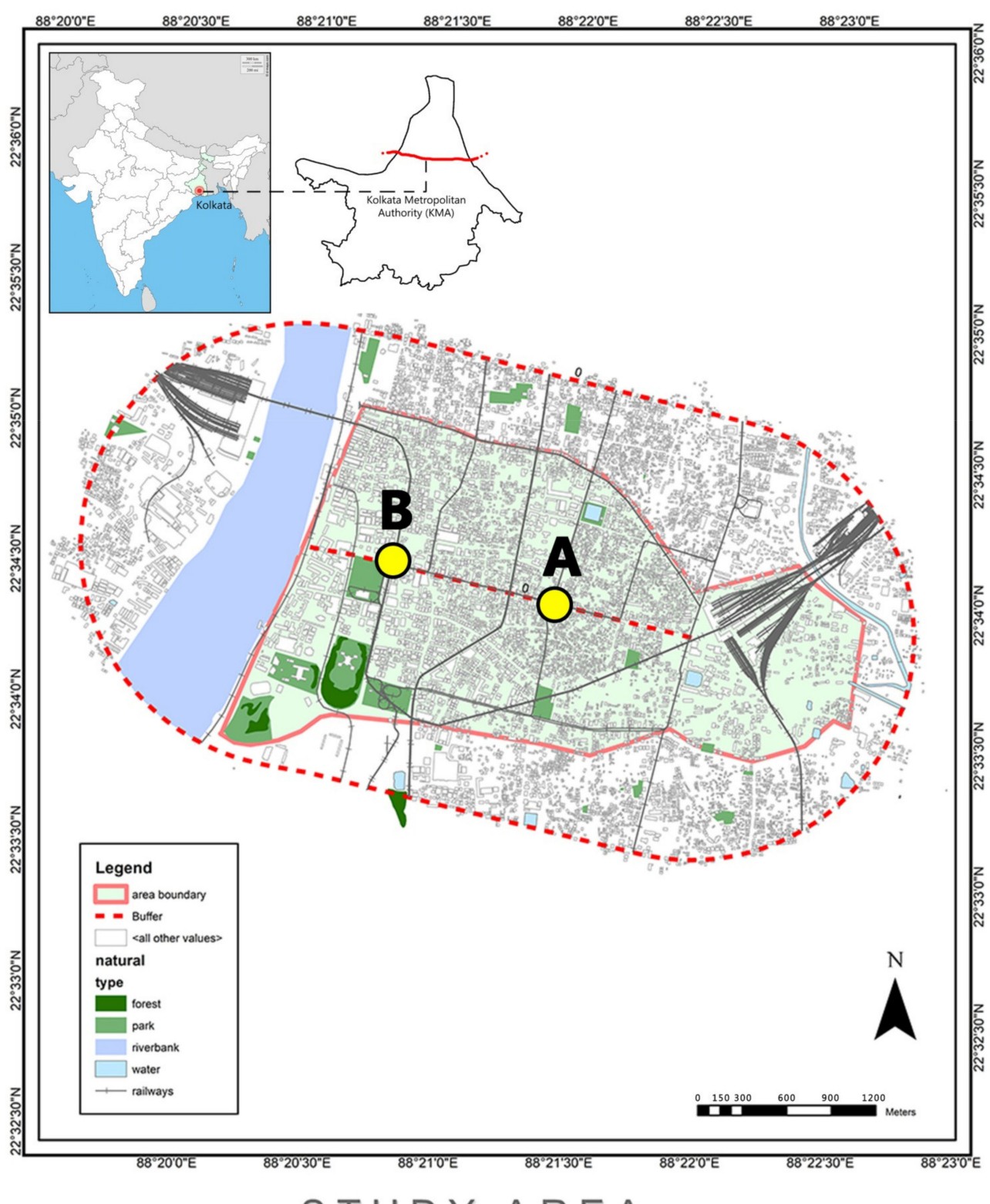

STUDY AREA

**Figure 18.** The location for the visual survey (between point A and B); map prepared in ARC-GIS (Source: Author).

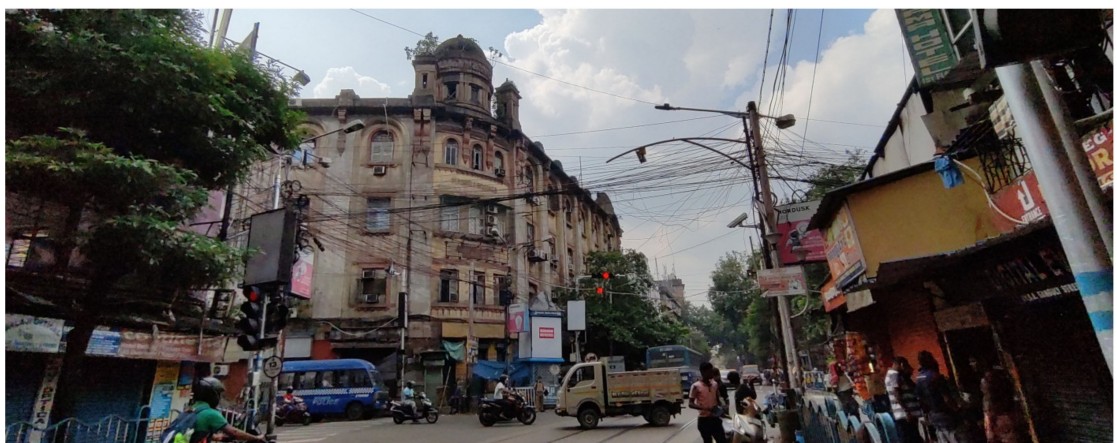

**Figure 19.** Mercantile Building (center) built in the early 1900s: an example of built heritage (Source: Author).

**Figure 20.** A cobbler occupying a footpath stretch along the external wall of Bowbazar Post Office (Source: Author).

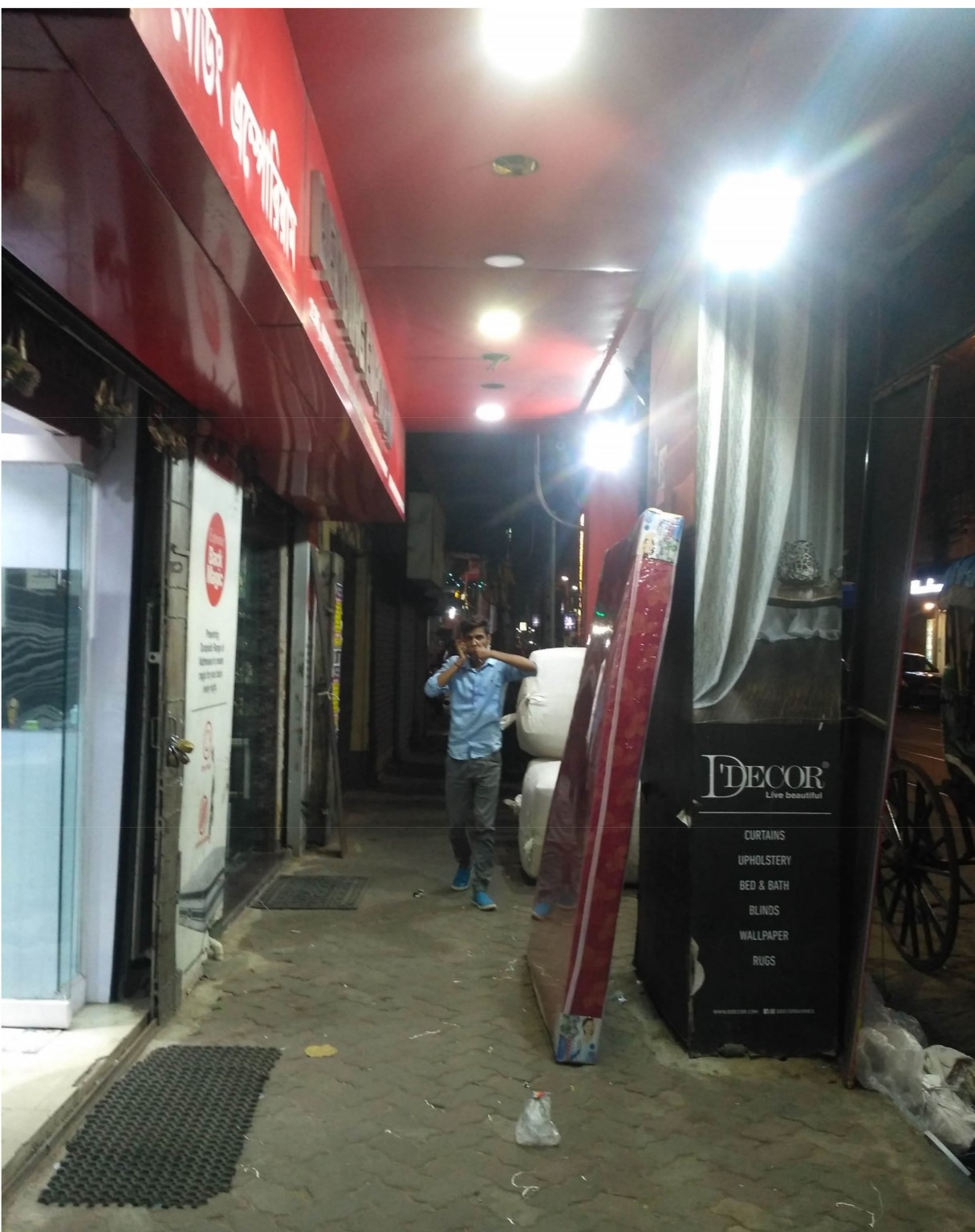

**Figure 21.** The shop on the left of the picture is using their products for display, on the pedestrian space (Source: Author).

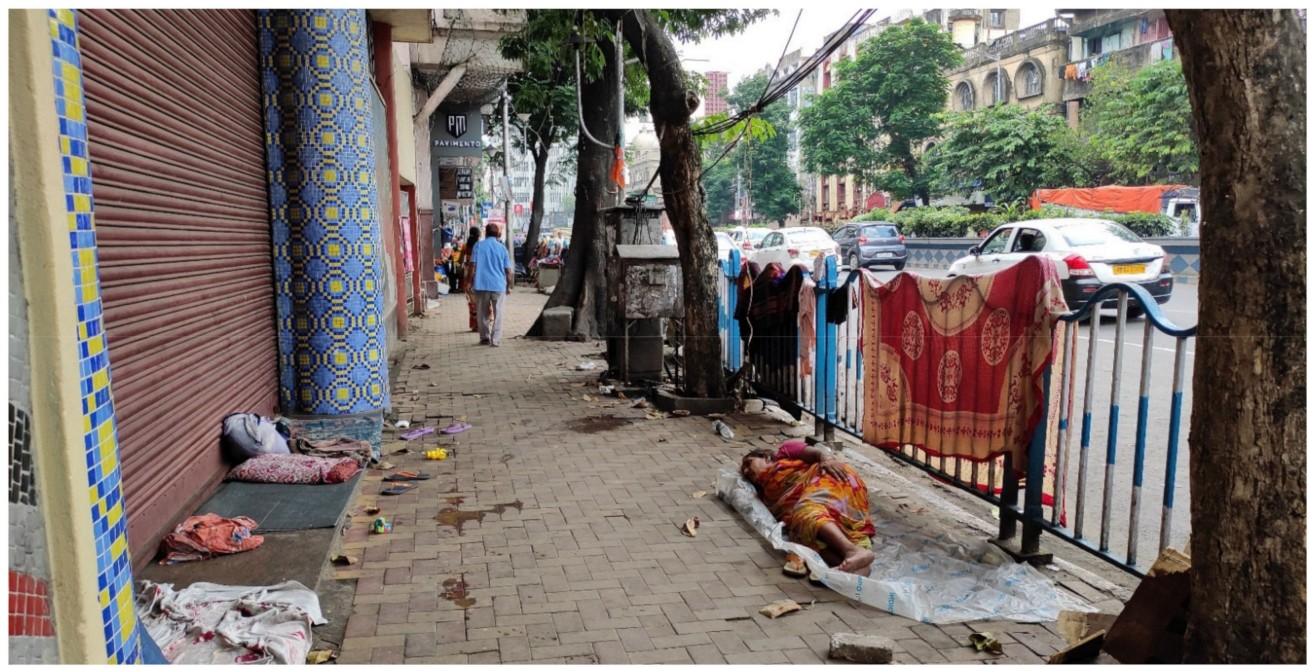

**Figure 22.** Homeless people sleeping on the streets (right) and also at the unused entrance of Gate 4 of Central Metro (Source: Author).

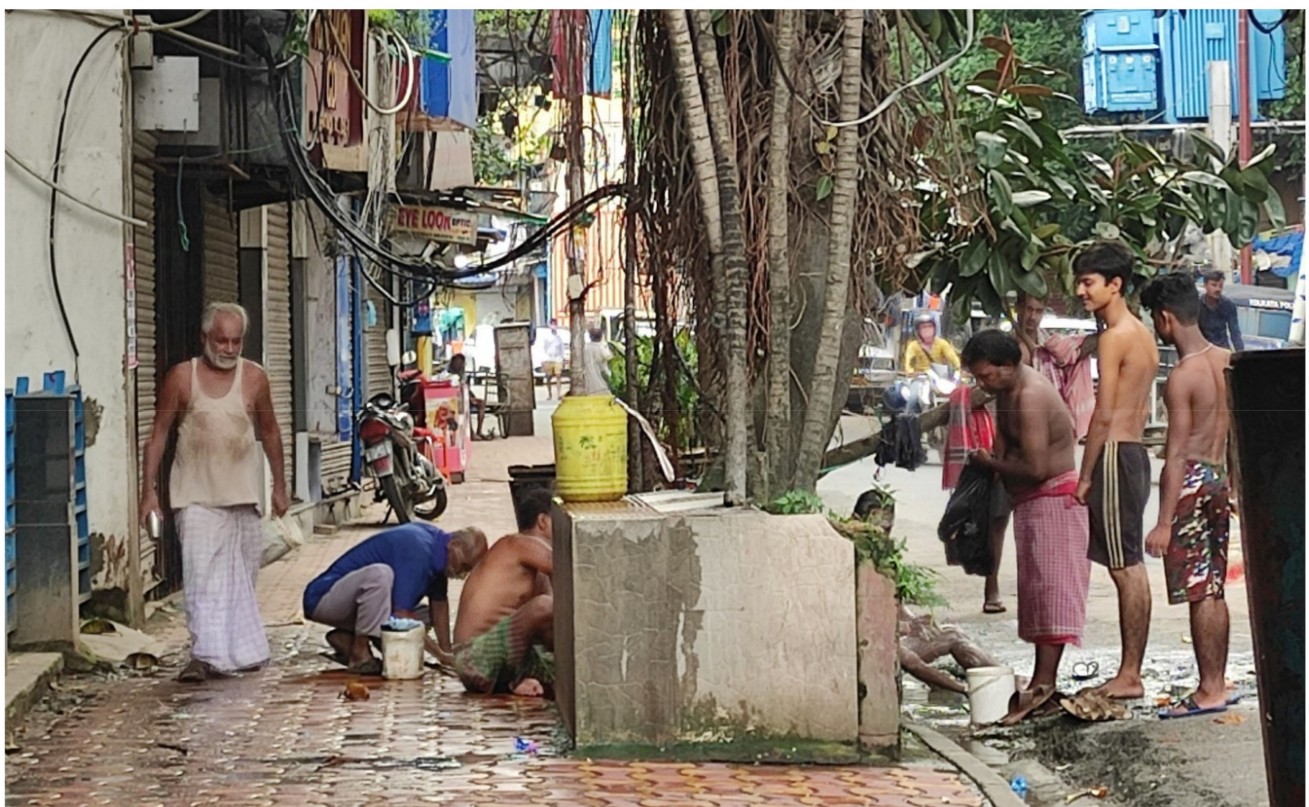

**Figure 23.** Open public bath still in operation at Bentinck Street, also capturing a part of the footpath as well as disturbing the vehicular stretch (Source: Author).

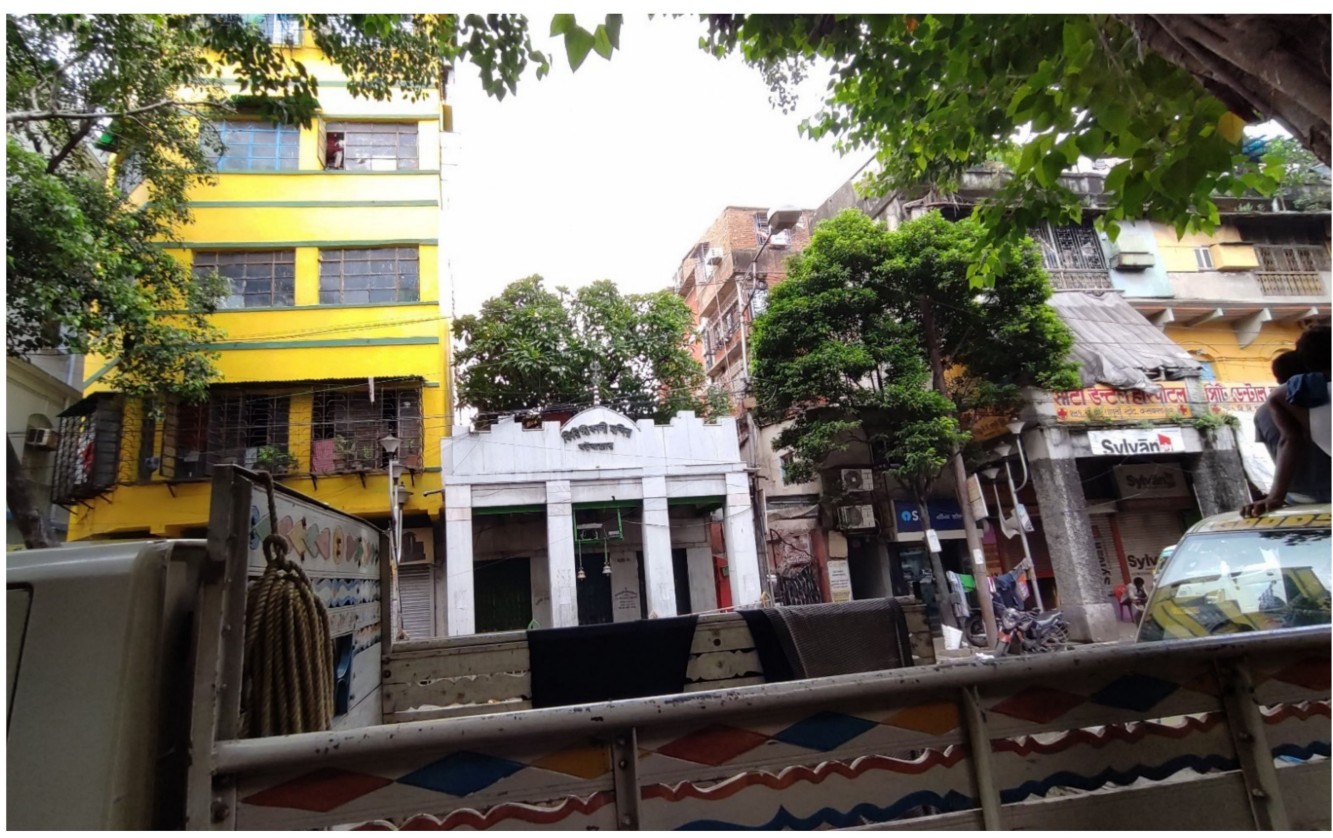

**Figure 24.** Almost 500-year-old temple "Firinghi Kali Bari": a place where primarily Hindus gather during prayer (Source: Author).

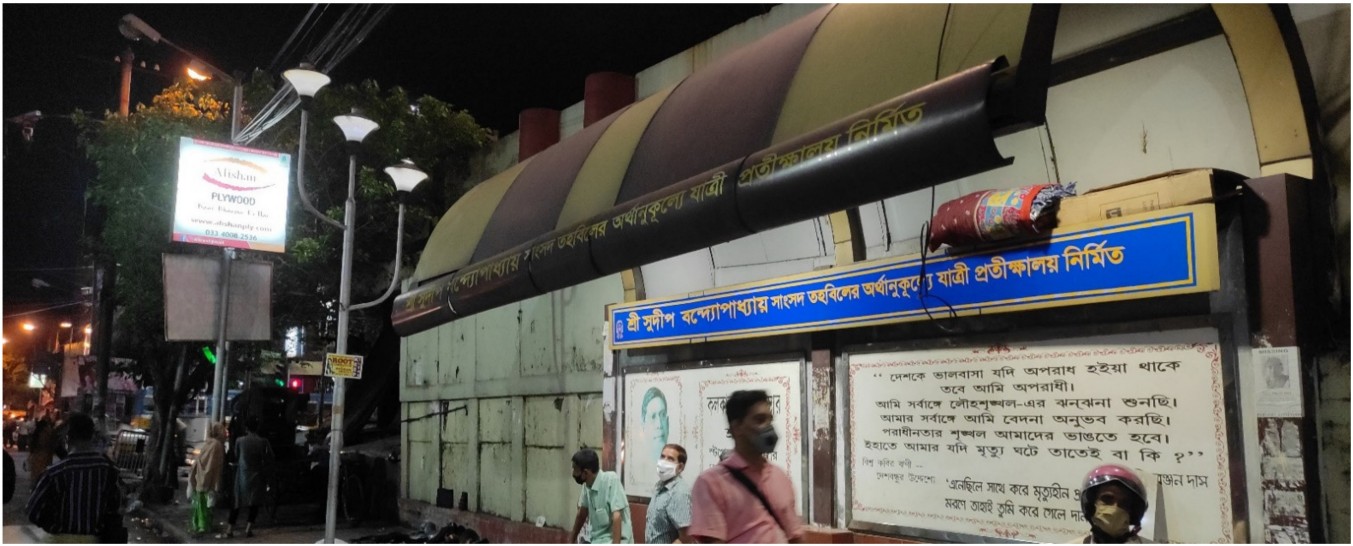

**Figure 25.** A bus stop between Central Avenue Crossing and Bow Street having bedding of homeless people stored above the seating area (Source: Author).

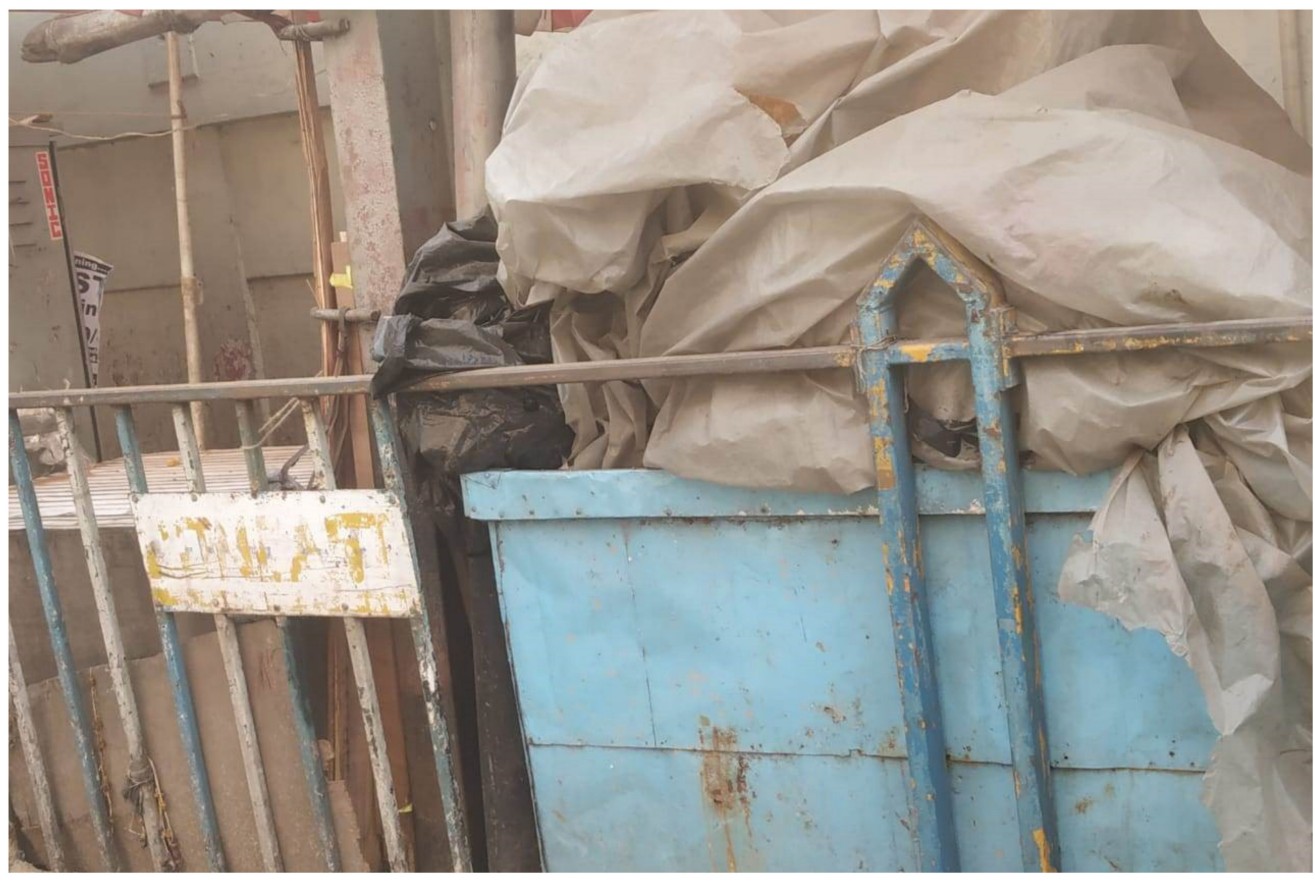

**Figure 26.** Railing near Gangadhar Babu Lane used by vendors as an edge for their storage (Source: Author).

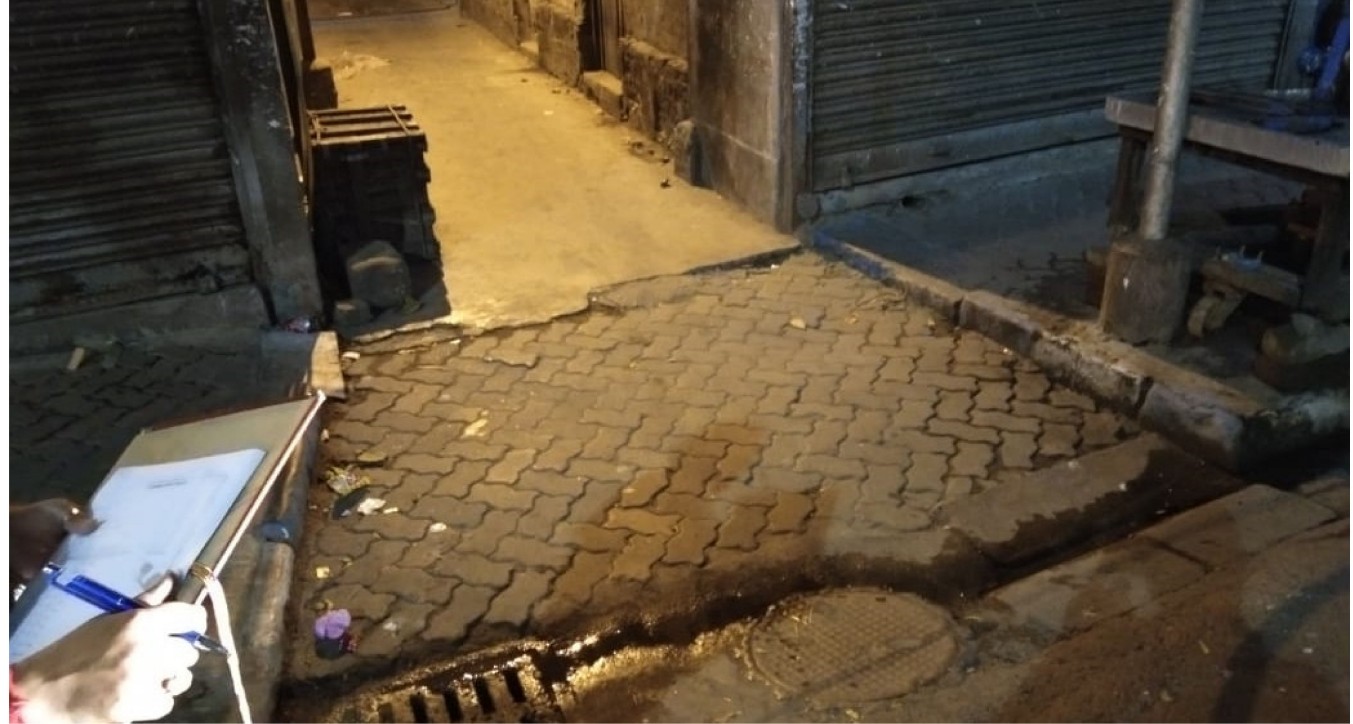

**Figure 27.** Entrance to Bibi Rozio Lane, showing poor kerb and drainage conditions (Source: Author).

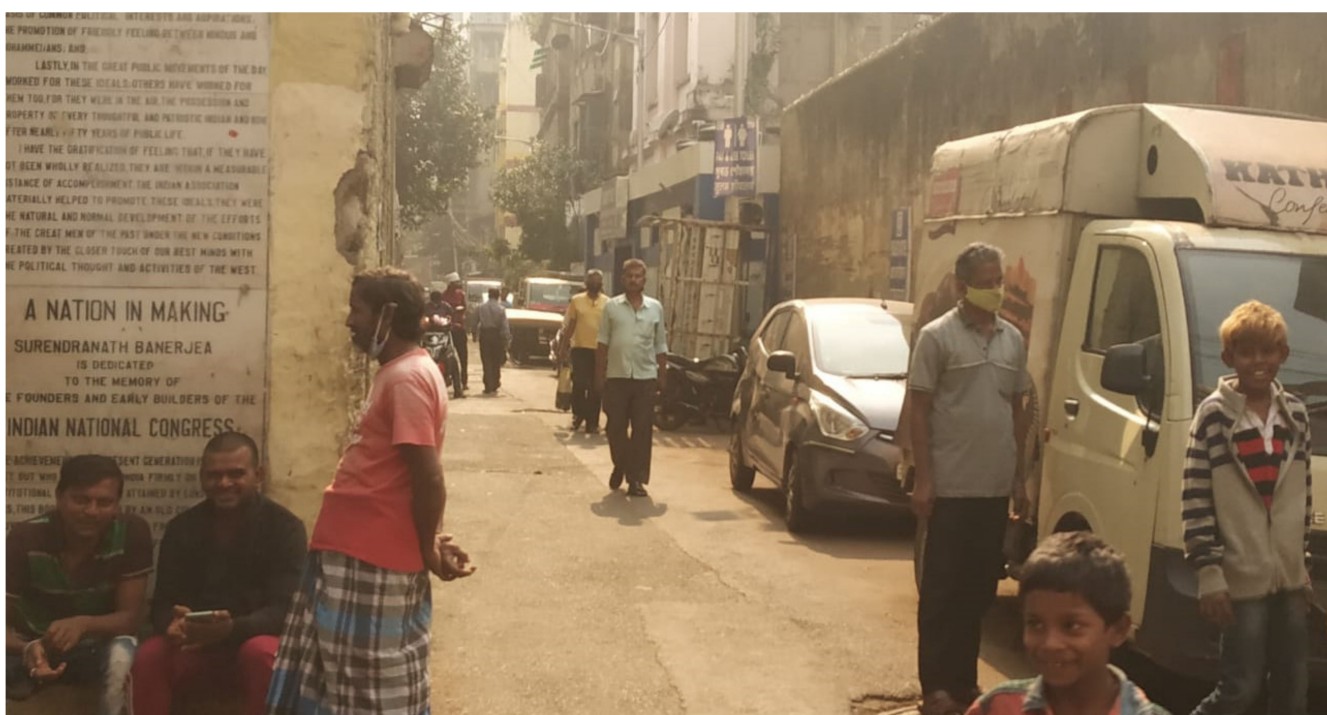

**Figure 28.** Informal parking of private and commercial vehicles consumes at least 25% of the available pedestrian space in Kenderdine Lane. Residents use extended plinths from heritage buildings as shaded seating space. (Source: Author).

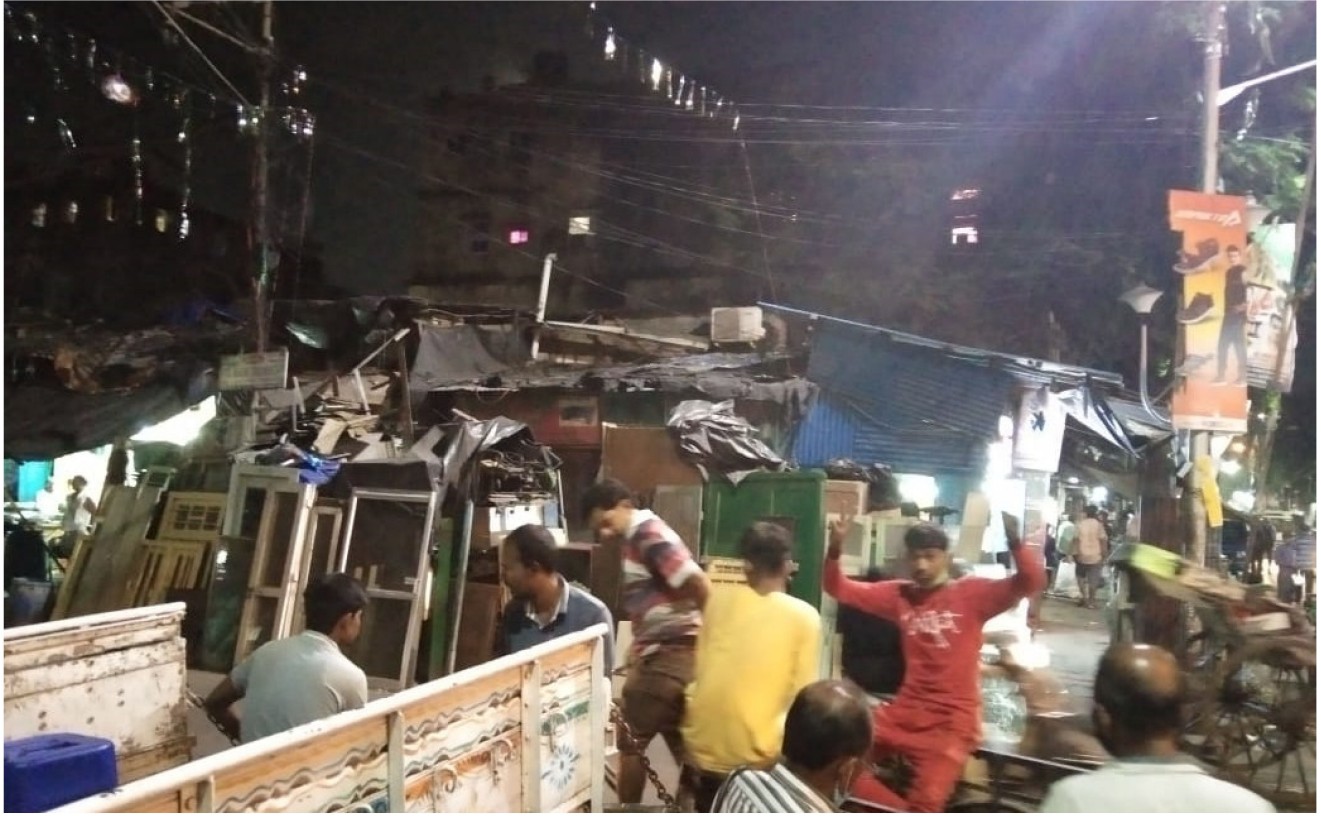

**Figure 29.** At The entrance of Phears Lane during evening peak hour, a multiplicity of transport modes, absence of signals and, improper pedestrian facilities creates a chaotic environment. (Source: Author).

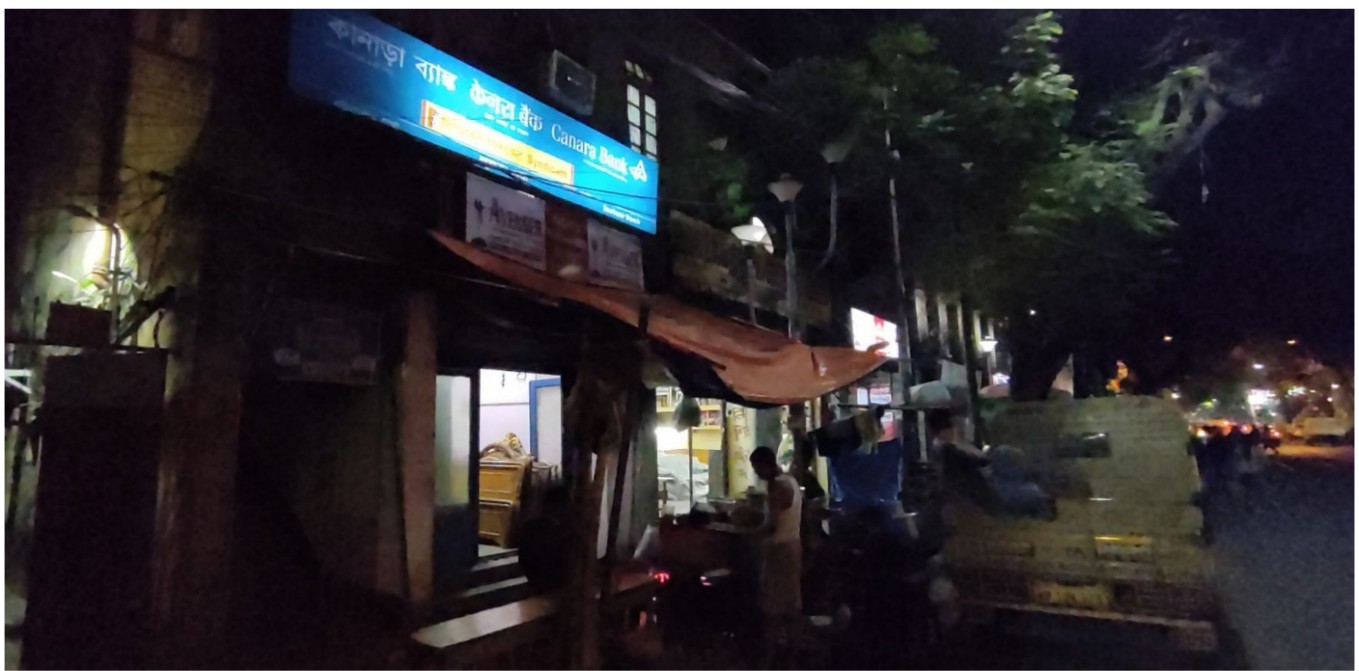

**Figure 30.** Poor lighting conditions in a stretch between New Bowbazar Lane and Kenderdine Lane (Source: Author).

### 3.3.2. Observation and Analysis

The distance between Point A and Point B (as shown in Figure 18) is 850 m, and it takes about fourteen minutes for an able-bodied person for covering the distance. The stretch is oriented in North-East to South-West direction and has eighteen junctions of different scale; based on the (a) width of the streets converging in the junction, (b) height of the buildings along the streets, and (c) predominant type of activities). The surveyed locations are Nirmal Chandra Dey Street, New Bowbazar lane, Kenderdine Lane, Central Avenue (GATE 4_Yogayog Bhawan), Central Metro (GATE 1_Indian Airlines), Bow Street, Metcalfe Street, Bentick Street, Rabindra Sarani Rd., Chatawalla Gully, Phears Bye Lane, Phears Lane, Giri Babu lane, Central Metro (GATE 2_Lalbazar), Central Avenue (GATE 3_RITES), Gangadhar Babu lane, Bibi Rozio Lane, and College Street. The authors conducted observational walks along this stretch and captured the photographs. The authors recorded observations at various intervals of the day at: (a) 1300 h, when school finishes, (b) 1800 h when offices close for the day, and (c) 2100 h when commercial establishments shut down for the day. The observations made by the authors are explained hereafter.

Unorganized informal vending and encroachment by existing establishments occupied nearly every street [refer to Figures 20 and 21, respectively]. Complemented by this, numerous beggar/homeless people were occupying a share of the pedestrian space (often near non-operating Metro rail gates) [refer to Figure 22]. Communal open baths, which were a scene from yesteryear, were also present in certain areas [refer to Figure 23]. A mix of Hindu, Muslim and Christian religious places were present in the stretch [refer to Figure 24]. In terms of transportation, the bus stops were in poor infrastructural condition [refer to Figure 25]. Informal vendors used the railings along the footpath for their personal use [refer to Figure 26]. Additionally, the kerb along the footpaths was in dilapidated condition [refer to Figure 27]. Informal parking was observed in almost the entire stretch [refer to Figure 28]. Lack of signals at crucial intersections created chaos, especially during peak hour traffic, as depicted in Figure 29. Poor functioning of street lights were observed at multiple locations such as the one photographed in Figure 30.

### 3.3.3. Findings and Discussion

The visual survey helped the authors in finding certain issues in basic mobility which requires thorough examination before interpreting Universal Mobility. Table 5 shows the

observations from this visual survey. "Universal Mobility" can be interpreted for this stretch only after assessing minimum mobility standards. Thus, these observations are the first stage for apprehending the mobility issues in this stretch.

**Table 5.** Learning from the visual survey in Bipin Behari Ganguly Street in Central Kolkata, India between September and November 2020 to assess Universal Mobility conditions (Source: Author).

| S.No | Topic | Inference |
|---|---|---|
| 1 | Predominant building use | The mixed land use and multiplicity of usage in a single building creates a complex structure of users. A different activity such as educational, business and others create multiple "peak hour traffic" which the present infrastructure is not adequate to cater. |
| 2 | Heritage buildings | A large number of Heritage buildings fostered low temporal change and is also restricting alteration in the width of carriageways. Typical heritage buildings with no setback and footpaths are not segregated from the building entrances creates a pedestrian discomfort in this stretch. |
| 3 | Informal vending | Since the inception of this stretch, informal vending has been a characteristic feature. However, with increasing population and vehicular pressure, the informal vendors are presently a threat to mobility. |
| 4 | Encroachment | Encroachment is a policy failure. Illegally occupied spaces within footpath create severe problems during peak hours. |
| 5 | Beggar/homeless/child labour | Beggar/homeless/child labour is a social issue. Only socio-political intervention can facilitate the process of emptying the streets of these user groups. |
| 6 | Communal open bath | Communal open baths are obsolete in most parts of the city except central part of Kolkata. These communal open baths (often at the edge of footpath and street) serve as an important functional social infrastructure due to the presence of many daily wage workers and floating population. On the contrary, the baths are posing threat to mobility on both footpath and street. The common baths create an extra crowd and spilt water, both of which are threatening to the pedestrian environment. |
| 7 | Religious establishments | Being an organically developed area within historic core, Hindu, Muslim and Christian crowds are proportionately present. The "Firinghi Kali Bari" Temple and "St. Xavier's Church" are examples of religious structures. The problems with these locations are their presence along the footpath and absence of alternative entry. So, during prayers hours, a conflict in pedestrian movement due to different user group at the same line of movement is observed. |
| 8 | Bus stop | The bus stops are in dilapidated condition and used by homeless people to store their belongings. Provision of information display is absent in bus stops; rather, the bus stops are used more for advertising purposes. A bus stop without facilities and improper information display is of seldom use in the 21st century. |
| 9 | Metro stations | The unused metro station gates are occupied by homeless people as their temporary shelter. These issues are to be solved at the policy and socio-political level. |
| 10 | Signalized intersections | Allocation of signals at intersections in this stretch was done decades ago in this stretch. The new zones of pedestrian and vehicular traffic have not been considered in the recent past. Due to different building use, the volume of traffic is varying. The concept of allocating signals based on road width might not be suitable for this area. Rather, signals based on traffic volume and predominant building uses are apt for this stretch. |
| 11 | Railing | Informal vendors occupied the railings in most parts of the stretch. Neither a clear demarcation for the pedestrian and vehicular traffic, nor scope for elderly/differently-abled people who needs to hold the railing, is present. A clear edge is essential for an ideal streetscape. |
| 12 | Kerb | Kerb which essentially provides gradation in street-level mobility is in extremely dilapidated condition in this stretch. As a result, neither able-bodied pedestrians nor differently-abled people are able to move freely in the footpaths. The scenario worsens at night due to lack of adequate light. |
| 13 | Storm water drains | Storm water circulation in this part of Central Kolkata is better than most of the parts; however, the location of storm water drains are a matter of major concern. The location, grating style and the slope is dangerous for pedestrian, especially the ones with walking cane. |

**Table 5.** *Cont.*

| S.No | Topic | Inference |
|------|-------|-----------|
| 14 | Signage | There was no directional signage in the entire stretch. Public utilities are not demarcated. The historic buildings had their description engraved near the entrance. However, other places of public interest are not having any information. Thus, for people new to this area and people with cognitive issues, they face difficulty in traversing the stretch. |
| 15 | Public Toilet, Drinking water facilities, and resting facilities | Only one public toilet was observed in the entire stretch of 850 m in which visual observation was undertaken and no drinking water facility was present. Public facilities such as drinking water and public toilets complemented by street furniture are components of a healthy street. Mobility without public facilities is unreasonable in the urban scenario. |
| 16 | Trash Bins | According to the latest "Swachh Survekshan" programme or Clean India Campaign launched in 2016, Trash bins are to be placed at fifty meters interval along urban streets. However, in this stretch, hardly any trash bins were observed and as a result, a littering was observed. Clean streets promote increased mobility and should be taken care of. |
| 17 | Street Lights | In spite of the fact that street lights are present all throughout the stretch, the requisite lighting intensity was not present. When checked using the "Light meter" application, most of the streets showed a lux level of less than ten, which is not ideal for a safe pedestrian environment. The fact that this stretch experiences a heavy pedestrian footfall, poor lighting makes it even more vulnerable to users with cognitive difficulties. |

## 4. Major Findings

In this section, the arguments and discussions in this research are summarized by establishing the relationship between the "objectives" and "survey undertaken" by the authors.

### 4.1. Finding 01

The first objective of this research paper (i.e., to ascertain the need for a new dimension in the Indian accessibility scenario) was substantiated through the first survey (as elaborated in Section 3.1) involving the design fraternity of India. Taking a cue from the first study about the focus area of accessibility in transportation, the next step was exploring the second objective (i.e., what is people's perception regarding mobility in an Indian old city?).

### 4.2. Finding 02

The second objective of this research paper (i.e., to assess people's perspective towards Universal Mobility in core urban areas in India) was verified through the findings from the survey involving the residents of Kolkata (as elaborated in Section 3.2). Once this scenario was clear, the next steps were choosing a particular stretch in the old core of Kolkata and conducting a visual observation which would explain a part or whole of people's opinions regarding mobility in an Indian old city.

### 4.3. Finding 03

The third objective of this research paper (i.e., to identify the issues in mobility in core urban areas in India) was advocated through the inferences from the visual survey of a stretch in the old core of Kolkata (as elaborated in Section 3.3). The observations and inferences through this visual survey indicated that mobility conditions are not in a positive state in this stretch. Basic infrastructural issues are in downtrodden conditions. Thus, reimagining this stretch with Universal Mobility concerns is a difficult task altogether. In order to initiate the process of Universal Mobility features in this stretch, however, a need for further analysis of this stretch was necessary.

Thus, it is inferred that the reluctance of people in old cities towards walkability and usage of public transport is directly linked to the poor infrastructural conditions. The historic origin of the old core in Indian cities and their organic pattern of development generate a chaotic urban scenario. Presently, dissatisfactory levels of cognition at street level, which is not an ideal scenario for able-bodied and differently-abled alike, are persistent. Elderly people additionally face problems due to this dilapidated state.

## 5. Conclusions

In spite of a number of policies and programs related to accessibility at the national level, Indian old cities are often ignored. The reason for this phenomenon is largely the Indian Constitution. According to the Seventh Schedule (Article 246) of the Indian Constitution, "Land" is a state subject. It implies that a decision regarding urban development is a matter of individual state [41]. Thus maintaining coherence with national policies is a political choice for the Chief Minister of a state. Kolkata is located in the state of West Bengal and as of January 2021, the state-ruling electoral party (Trinamool Congress with Smt. Mamata Banerjee as the state's Chief Minister) is not in alliance with the central-level ruling party (Bharatiya Janata Party, with Shri. Narendra Modi as the country's Prime Minister). Thus, it was not unnatural when 92.16% of the respondents from elderly/differently-abled and 85.14% of able-bodied people responded during the survey that they have not heard about the Accessible India Campaign.

Thus, identifying a custom-made access audit format specifically for a particular stretch shall be beneficial for the old core cities, in terms of its practical applicability by avoiding political complications. At the same time, it will also serve an altered guideline for old cities, rather than the generic national guideline. A total of 63% of respondents from the architecture fraternity during the survey mentioned the same suggestion.

As responsible professionals in the field of Architecture and Planning, the authors are taking part in the movement of creating a built environment "For All". Built environment "For All" takes into attention the needs of able-bodied as well as elderly/differently-abled people. The role of the Laboratory of Architectural Planning, Hokkaido University under the supervision of Professor Dr. Mori and Associate Professor Dr. Nomura is specifically noteworthy in this discussion. This laboratory focuses on "planning architecture" based on "practical problem interest". One of the research themes of this laboratory is "research on the ideal living environments for minorities", with a sub-theme titled "Environment design that realizes safe and comfortable going out for the physically vulnerable". This enabled the authors to conduct this research in this laboratory. Moreover, the focus of this laboratory is also in coherence with Goal Number 11 of United Nations Sustainable Development Goals (UN-SDG). The title of the goal "Sustainable cities and communities" mentions the need to provide (a) access to safe, affordable, accessible, and sustainable transport systems for all and to (b) improve road safety, notably by expanding public transport, with special attention to the needs of those in vulnerable situations, women, children, persons with disabilities and older persons. This goal indicates the need for Universal Design and Universal Mobility, in a global society. In 2020, Hokkaido University collaborated with twenty-seven other Japanese Universities for participation in United Nations University "Sustainable Development Goals University Platform" (SDG-UP). This platform established by the UN-University-Institute for the Advanced Study of Sustainability (UNU-IAS) emphasizes global development in sustainable terms. Thus, "Laboratory of Architectural Planning" can prove an essential resource body in assessing the Universal Mobility conditions in old core cities by aligning its research interest with global concerns in the field of Universal Design.

Capacity building and research on Universal Design is an effective tool for creating global awareness about the need of Universal Design in Architecture and Planning [42]. Along similar lines, the Ministry of Housing and Urban Affairs, Government of India, in collaboration with- IIT (Indian Institute of Technology) Roorkee, NIUA (National Institute of Urban Affairs) and AIILSG (All India Institute of Local Self Governance), has launched an initiative titled BASIIC (Building Accessible, Safe, and Inclusive Indian Cities). Functionally started in 2020, the BASIIC project is supported by the DFID (Department for International Development, now known as FCDO or Foreign, Commonwealth and Development Office fund of the UK government. This project aims towards framing disabled-friendly guidelines and policy recommendations that can be implemented through the already functioning Smart Cities Mission [43]. The major objectives of the project include (a) training and capacity building of government officials and (b) sensitizing citizens of India about the

importance of Universal Design. Eminent architects and planners are engaged by NIUA and AIILSG in this project, from the initial stage of preparing posters for Universal Design awareness until final preparation of the course modules. The first author is presently engaged as the content curator and one of the module development experts for this project. Unlike previous projects of Government of India, BASIIC project focuses on the concept of spreading awareness on Universal Design through education and research. Thus, in the coming years, accessibility conditions might improve in urban areas of India.

The authors believe that sharing knowledge on Universal Design through research and interaction increases the possibility for an inclusive society. Such a society shall ensure the same standard to urban infrastructure for both able-bodied and differently-abled. However, capacity building alone cannot bring about a radical change in critical Universal Design thinking. Thus, the authors state that the findings for this paper are comparatively the beginning of a quest towards assessing accessibility in the core city areas in Indian context.

For further research in the same domain, a checklist-based assessment could be undertaken by interested researchers. This checklist, as mentioned in the aims for this paper, shall contain the factors for an ideal accessibility audit checklist which is to be used in old core Indian cities. Further research could be based on three distinct lines of action. The three lines of action are a) Universal Mobility features, b) cognitive factors, and c) traffic volume. The last survey in this research (as detailed in Section 3.3) titled "Visual Observation of an Old City Core" which was undertaken in Bipin Behari Ganguly Street in Kolkata surfaced issues related to the infrastructural conditions in old core cities in India [Refer to Figures 19–30]. First, based on these issues, an in-depth analysis of each street and footpath stretches (space between the junction of a street and another) could be conducted for determining the infrastructural level in a quantitative manner. Second, the cognitive factors could be examined and thereby used for determining the linkage between five senses and pedestrian behavior. However, these two types of study will be incomplete if these studies are not correlated to the traffic volume. The traffic volume, consisting of pedestrian and vehicular volume shall help in assessing the level of service for the study stretch. Figure 31 illustrates the methodology that could be adopted for further study based on the findings of this paper. This further research based on the findings of this paper will foster the preparation of a rating system for Universal Mobility in the core of old Indian cities.

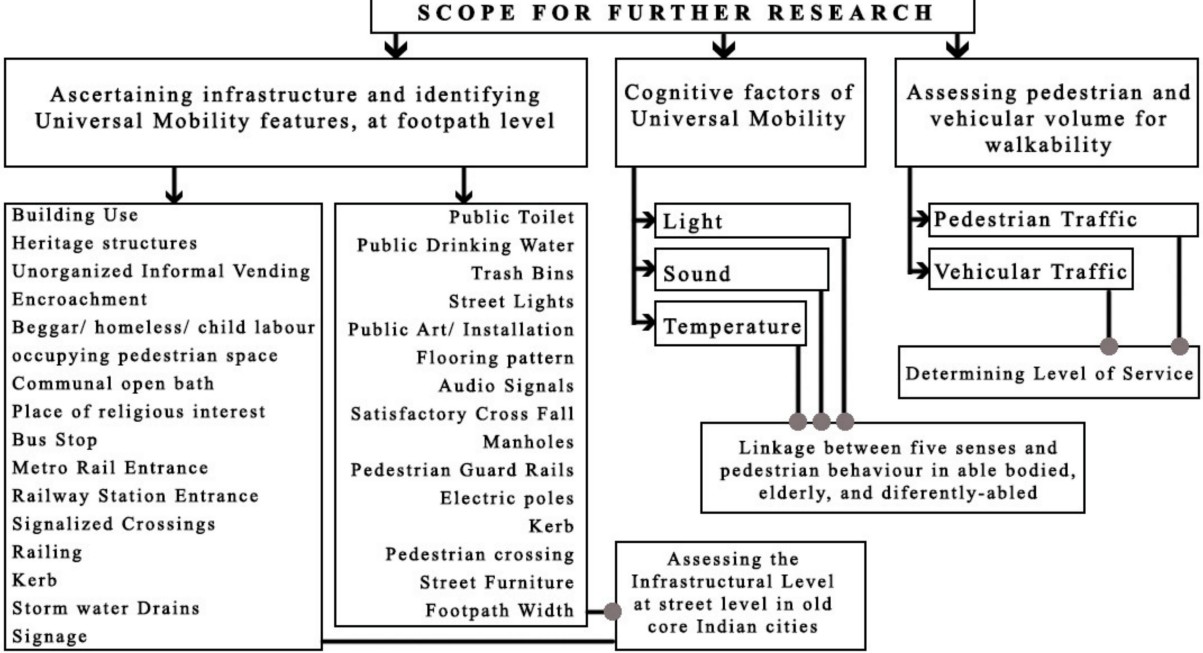

**Figure 31.** Methodology showing scope of further research based on the findings from this paper. (Source: Author).

Thus, the answer to the research question for this paper, i.e., "to find out whether core areas of urban India can be made inclusive in terms of accessibility" is "yes". Inclusivity can prevail in the old core of Indian cities provided a methodical approach towards Universal Mobility is practiced, as elaborated in this paper.

**Author Contributions:** Conceptualization, S.M. and G.D.M.; methodology, G.D.M.; software, G.D.M.; validation, S.M. and R.N.; formal analysis, G.D.M.; investigation, G.D.M.; resources, S.M., R.N. and G.D.M.; data curation, G.D.M.; writing—original draft preparation, G.D.M.; writing—review and editing, S.M. and R.N.; visualization, G.D.M.; supervision, S.M. and R.N.; project administration, S.M. and R.N.; funding acquisition, S.M. All authors have read and agreed to the published version of the manuscript.

**Funding:** This research received no external funding.

**Institutional Review Board Statement:** Ethical review and approval were waived for this study, since no data and information related to the ethical guidelines were at the discretion of the committee at Hokkaido University.

**Informed Consent Statement:** Informed consent was obtained from all subjects involved in the study.

**Data Availability Statement:** Not applicable.

**Acknowledgments:** The authors gratefully acknowledge the help of the people who accompanied them during survey in Kolkata.

1.  Manorama Mahanti: Retired Municipal Corporation Employee and Resident of study area [Contact: +91-6297943706]
2.  Disha Maity: Fourth Year Student in Bachelor of Architecture course [Contact: dishamaiti@gmail.com]
3.  Akash Das: Economics Graduate and Social Worker [Contact: dasakash0710@gmail.com]
4.  Sagnik Das: Fifth Year Student in Bachelor of Architecture course [Contact: sagnikdas565@gmail.com]

The authors, especially the first author, are also thankful to the people from the design fraternity of India who helped in the initiative of workshops on Universal Design in India between 25 June 2020 and 8 November 2020.

1.  Sandeep Pathe: Principal Architect, Studio Sakha
2.  P Satheesh Kumar: Professor and Dean, School of Architecture and Interior Design, SRM Institute of Science and Technology, Kattankulathur
3.  Anjali S. Patil: Associate Professor, Department of Architecture, Madhav Institute of Technology and Science, Gwalior
4.  S. Kumar: Professor and Principal, Jawaharlal Nehru Architecture and Fine Arts University, Hyderabad
5.  Uma Jadhao: Professor and Principal, School of Architecture, DY Patil University, Pune
6.  B. Rajendra Koli: Professor and Principal, Akhil Bharatiya Maratha Sikshan Parishad's Anantarao Pawar College of Architecture, Pune
7.  Neeraj Gupta: Professor and Head, School of Architecture, Central University, Ajmer
8.  Uma Ranganathan: Professor and Principal, Rajalakshmi School of Architecture, Mevalurkuppam
9.  Vasanti Londhe: Professor and Principal, Marathwada Mitramandal's Institute of Environment and Design's College of Architecture, Pune
10. Tanya Gupta: Professor and Head of the Department, School of Architecture, Delhi Technical Campus, Noida
11. Dipankar Sinha: Retired Director General, Town Planning Division, KMC

Lastly, the authors acknowledge the suggestions of Rachna Khare (Dean, School of Planning and Architecture) for explaining the evolution of Universal Design in the field of urban mobility

**Conflicts of Interest:** The authors declare no conflict of interest.

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
