# Peer review of "Universal Mobility in Old Core Cities of India: People’s Perception"

_sustainability, doi:10.3390/su13084391_

Round 1

Reviewer 1 Report

The manuscript focuses on a very interesting topic.
The text contains several grammatical errors.
A flow chart should be included in the introduction explaining the steps of the research carried out and paragraph 1.2 should be deleted.
It is considered appropriate to include more literature related to the concept of accessibility in its various definitions and limitations, therefore it is possible to refer to the following works :

1) Mrak, I., Campisi, T., Tesoriere, G., Canale, A., & Cindrić, M. (2019, December). The role of urban and social factors in the accessibility of urban areas for people with motor and visual disabilities. In AIP Conference Proceedings (Vol. 2186, No. 1, p. 160008). AIP Publishing LLC.
2)Campisi, T., Ignaccolo, M., Inturri, G., Tesoriere, G., & Torrisi, V. (2020, July). The Growing Urban Accessibility: A Model to Measure the Car Sharing Effectiveness Based on Parking Distances. In International Conference on Computational Science and Its Applications (pp. 629-644). Springer, Cham.
3)Rahman, M. T., & Nahiduzzaman, K. (2019). Examining the walking accessibility, willingness, and travel conditions of residents in Saudi cities. International journal of environmental research and public health16(4), 545.

4)Canale, A., Campisi, T., Tesoriere, G., Sanfilippo, L., & Brignone, A. (2020, July). The Evaluation of Home-School Itineraries to Improve Accessibility of a University Campus Trough Sustainable Transport Modes. In International Conference on Computational Science and Its Applications (pp. 754-769). Springer, Cham.

5)Basbas, S., Mintsis, G., Oikonomou, K., & Taxiltaris, C. (1970). Design of a sustainable and accessible environment in central areas. WIT Transactions on Ecology and the Environment84.

More explanation of Table 1 is required.
We do not understand the purpose of the research as well as the following rows and figures 3 and 4.
It is considered appropriate to 
1) the inclusion of research limitations
2) definition of the research as a universal methodology or exclusive to the case examined 
3) inclusion of future research steps

The numbering of the literature is not homogeneous and does not conform to the formatting 

Author Response

Respected Reviewer, we sincerely thank you for your kind comments and for providing a platform to modify and strengthen the research paper.

Your comments were taken up by us and necessary modifications were made and enlisted below. The RESPONSE TO YOUR KIND COMMENTS along with MODIFIED MANUSCRIPT has been attached herewith.

Reviewer 2 Report

The paper “Universal Mobility in old core cities of India:  People’s Perception” is focusing on an interesting topic however the authors must rewrite and reorder the structure of the paper.

Here are my comments:

Introduction: rewrite the introduction considering the relevant international literature on this topic. Include the research gap; this should be related to the international literature review.

Methods: Page 4 Include methods. Please describe the methods used and your rationale for their use.

Page 4 Literature: this is a long section; delete it and some information can be part of your methodology.

Ethical issues must be addressed within the paper.

Results: it needs major rewriting! Results should be clear and concise. Include only relevant results.

Visual observation of an old city core: This part of your methods.

Discussion: discuss your findings with previous research. Include statistical significance.

Conclusion: Highlight the importance and originality of your research.

Author Response

(The authors gave the same response as above.)

Reviewer 3 Report

Authors state that the design fraternity in India suggests the need of separate accessibility guidelines for old and new cities in India. However, along the paper it is not clear the specific difference between a modern and an old city, apart the reference to the heritage buildings in the Table 5. The particular reference to an old city requires a more in-depth disquisition about this aspect.

Authors do not give any indication about technical contents of the Access Audit Checklist through Architectural Planning, which is the first step in proposing a framework for Universal Mobility.

Author Response

Respected Reviewer, we sincerely thank you for your kind comments and for providing a platform to modify and strengthen the research paper.

Your comments were taken up by us and necessary modifications were made and enlisted below. The RESPONSE TO YOUR KIND COMMENTS along with MODIFIED MANUSCRIPT has been attached herewith.

[Please see the attachment]

Round 2

Reviewer 1 Report

The manuscript still contains grammatical and formatting errors.
Please check what is written on line 225.....probably a misprint.
On line 352 the correct bibliographic reference is "Mrak, I., Campisi, T., Tesoriere, G., Canale, A., & Cindrić, M. (2019, December). The role of urban and social factors in the accessibility of urban areas for people with motor and visual disabilities. In AIP Conference Proceedings (Vol. 2186, No. 1, p. 160008). AIP Publishing LLC."
The surname of one of the authors is Canale and not Antonino.
Formatting of page 15 needs revision as there are blanks and figures are too large. 
Tables should be reduced in their content 
The figure on page 23 should not be inserted as a full-page spread.
The manuscript should be re-organised to make it more compact and to include the main figures as much as possible, leaving the rest to the appendix.

The title references should be included in the bibliographical part.

Author Response

Dear Reviewer,

Thank you for your comments as mentioned below. The response (in Green) is attached herewith.

1) Please check what is written on line 225.....probably a misprint.

Response: MODIFIED from "Figure 1 summarizes the Research Process undertaken for this research" to "Figure 1 summarizes the process undertaken for this research"

2) On line 352 the correct bibliographic reference

Response: MODIFIED

3) The surname of one of the authors is Canale and not Antonino.

Response: MODIFIED

4) Formatting of page 15 needs revision as there are blanks and figures are too large. 

Response: TO BE ADJUSTED DURING FINAL COMPOSING BY PUBLISHER

5) Tables should be reduced in their content 

Response: The contents are not altered as they contain the requisite summary

6) The figure on page 23 should not be inserted as a full-page spread.

Response: Depending on the scale of the Map, the size is satisfactory.

7) The manuscript should be re-organised to make it more compact and to include the main figures as much as possible, leaving the rest to the appendix.

Response: TO BE ADJUSTED DURING FINAL COMPOSING BY PUBLISHER.

Reviewer 2 Report

Page 3: limitations are part of your discussion.

Page 4: 1.2 Research Process- Include methods!

Page 9: write Results, not  Survey and Results

Author Response

Respected Reviewer, we sincerely thank you for your kind comments and for providing a platform to modify and strengthen the research paper. Your comments were taken up by us and comments are mentioned below.

The comments from all 3 reviewers are incorporated in the paper and based on the context, content and structure of the paper the following responses are mentioned:

1) Limitation is mentioned and that is apt for a paper of this genre.

2) As per previous comments, the Research process is already elaborated in [1.2 Research Process]. Any more description shall make the paper unnecessarily long.

3) For this paper, it is adequate to write Results with Survey. All the information that is required by the readers is provided at the requisite locations. 

Other than this, some grammatical changes are done and the modified paper is uploaded to the portal.

Round 3

Reviewer 1 Report

The manuscript still has formatting and grammatical errors.
Although corrections have been made, the manuscript is still difficult for readers to interpret. In my opinion the paper is too long and tiresome to read until the end. Probably many tables still need to be summarized and some images should be eliminated or introduced in the appendix (if really necessary).

After these changes, the paper will probably be ready for publication.

Author Response

Respected Reviewer, we sincerely thank you for your kind comments and for providing a platform to modify and strengthen the research paper. Your comments were taken up by us and comments are mentioned below.

The comments from all 3 reviewers are incorporated in the paper and based on the context, content and structure of the paper the following responses are mentioned:

1) The formatting is unclear since the manuscript is in the correction phase. Both the original manuscript along with their corrections are present in the paper. After the review is complete, the paper will be modified with respect to formatting. 

2) The grammatical errors have been further rectified. Still, if you find further corrections, we would be obliged if you could exactly specify the lines.

3) As per the suggestion of Reviewer 2, the paper has been elaborated. Around 1100 words were added to the paper, in the 'Research Process' section.

4) Tables and Figures are all important based on the context of the research.

5) In case the reader feels that the analysis is elaborated, they can refer to the tables and figures for summarised reading. 

Thank you.

The modified paper is uploaded to Sustainability Portal.

Reviewer 2 Report

the paper has been improved. Please find attached your manuscript with my suggestions.

Author Response

Respected Reviewer,

Thank you for your comments. The modifications and clarifications are enlisted below:

1) Line 151- All possible details have been uploaded. No further text material is possible to be added.

2) Line 223- Figure 01 is apt for the paper. No further modification is possible at this stage as it clearly defines the course of the research.

3) Line 511- As mentioned to other reviewers, YES we have permission to use their identity. 

4) Line 1037- We have replaced the term from "works cited" to "references". However, technically, "works cited" is more apt than "references".

5) Line 1128- In technical papers, there are 2 categories of literature. "works cited/ references" and "bibliography". The former refers to the in-line citations and the ones which have been categorically referred to. The latter refers to the sources used for general understanding and overall structuring of the paper. It is to be noted that the Bibliography was further updated to include another source from the First Reviewer. The detailed difference between the two can be understood from the following sources:- 

1st Source (New Mexico State University) - https://grants-nmsu.libguides.com/c.php?g=836275&p=5976701#:~:text=In%20Works%20Cited%20and%20References,cited%20the%20work%20or%20not. 

A “Works Cited” list is an alphabetical list of works cited, or sources you specifically called out while composing your paper. All works that you have quoted or paraphrased should be included. A “Reference List” is very similar to a Works Cited list. Bibliographies, on the other hand, differ greatly from Works Cited and References lists. In Works Cited and References, you only list items you have actually referred to and cited in your paper. A Bibliography, meanwhile, lists all the material you have consulted in preparing your essay, whether you have actually referred to and cited the work or not. This includes all sources that you have used in order to do any research.

2nd Source- https://www.bibliography.com/how-to/apa-references-works-cited-and-bibliography-differences/

What’s the Difference: Bibliography vs. Reference List?

A bibliography, on the other hand, is a list of all the sources you consulted to write your paper. Even if you did not use them directly in your paper, you’ll still list them in your bibliography. This is a key difference between works cited and bibliography. Keep this in mind when you’re deciding between using a bibliography vs. a reference list for your paper.

The updated version is attached herewith
